# Characterising plasmacytoid and myeloid AXL+ SIGLEC-6+ dendritic cell functions and their interactions with HIV

**Freja A. Warner van Dijk[1,2], Orion Tong[1], Thomas R. O'Neil[1,2], Kirstie M. Bertram[1], Kevin Hu[1,2], Heeva Baharlou[1,2], Erica E. Vine[1,2], Kate Jenns[1,2], Martijn P. Gosselink[3], James W. Toh[1,2,3], Tim Papadopoulos[4], Laith Barnouti[4], Gregory J. Jenkins[5], Gavin Sandercoe[6], Muzlifah Haniffa[7,8,9], Kerrie J. Sandgren[1,2], Andrew N. Harman[1,2], Anthony L. Cunningham[1,2ᴑ] *, Najla Nasr [1,2ᴑ] ***

**1** The Westmead Institute for Medical Research, Centre for Virus Research, Westmead, Australia, **2** The University of Sydney, School of Medical Sciences, Faculty of Medicine and Health, Sydney, Australia, **3** Department of Colorectal Surgery, Westmead Hospital, Westmead, Australia, **4** Department of Plastic, Reconstructive, and Aesthetic Surgery, Westmead Private Hospital, Westmead, Australia, **5** Department of Obstetrics and Gynaecology, Westmead Hospital, Westmead, Australia, **6** Department of Plastic Surgery, Norwest Private Hospital, Bella Vista, Australia, **7** Wellcome Sanger Institute, Wellcome Genome Campus, Hinxton, Cambridge, United Knigdom, **8** Biosciences Institute, Newcastle University, Newcastle, United Knigdom, **9** Department of Dermatology and NIHR Newcastle Biomedical Research Centre, Newcastle Hospitals NHS Foundation Trust, Newcastle upon Tyne, United Knigdom

ᴑ These authors contributed equally to this work.

* tony.cunningham@sydney.edu.au (ALC); najla.nasr@sydney.edu.au (NN)

**Data Availability Statement:** All relevant data are within the manuscript and its Supporting information files.

## Abstract

AXL+ Siglec-6+ dendritic cells (ASDC) are novel myeloid DCs which can be subdivided into CD11c+ and CD123+ expressing subsets. We showed for the first time that these two ASDC subsets are present in inflamed human anogenital tissues where HIV transmission occurs. Their presence in inflamed tissues was supported by single cell RNA analysis of public databases of such tissues including psoriasis diseased skin and colorectal cancer. Almost all previous studies have examined ASDCs as a combined population. Our data revealed that the two ASDC subsets differ markedly in their functions when compared with each other and to pDCs. Relative to their cell functions, both subsets of blood ASDCs but not pDCs expressed co-stimulatory and maturation markers which were more prevalent on CD11c+ ASDCs, thus inducing more T cell proliferation and activation than their CD123+ counterparts. There was also a significant polarisation of naïve T cells by both ASDC subsets toward Th2, Th9, Th22, Th17 and Treg but less toward a Th1 phenotype. Furthermore, we investigated the expression of chemokine receptors that facilitate ASDCs and pDCs migration from blood to inflamed tissues, their HIV binding receptors, and their interactions with HIV and CD4 T cells. For HIV infection, within 2 hours of HIV exposure, CD11c+ ASDCs showed a trend in more viral transfer to T cells than CD123+ ASDCs and pDCs for first phase transfer. However, for second phase transfer, CD123+ ASDCs showed a trend in transferring more HIV than CD11c+ ASDCs and there was no viral transfer from pDCs. As anogenital inflammation is a prerequisite for HIV transmission, strategies to inhibit ASDC recruitment into inflamed tissues and their ability to transmit HIV to CD4 T cells should be considered.

**Funding:** This work was funded by Investigator Grant APP1177942 awarded to ALC by the National Health and Medical Research Council (NHMRC, https://www.nhmrc.gov.au) and Philanthropic Funds by the Neil and Norma Hill Foundation to NN. ALC received a salary from the Investigator Grant and FW, EV, KJ are supported by research training program at the University of Sydney. The funders had no role in study design, data collection and analysis, decision to publish, or preparation of the manuscript.

**Competing interests:** The authors have declared that no competing interests exist.

## Author summary

This study highlights the significance of AXL[+] Siglec-6[+] dendritic cells (ASDC) in HIV transmission, particularly in inflamed peripheral tissues such as anogenital tissues, where HIV transmission is prevalent. It reveals that ASDCs are present in inflamed human tissues, including psoriasis affected skin, colorectal cancer and anogenital tissues. However, they are absent in uninflamed tissues. Furthermore, we investigated the expression of chemokine receptors that facilitate ASDC migration from blood to inflamed tissues, and their interactions with HIV and CD4 T cells. Notably, different subsets of ASDCs exhibit different expression levels of HIV binding receptors and showed trends of different phases of HIV transmission to T cells. Understanding ASDC involvement in HIV transmission could provide valuable insights for developing strategies to inhibit their recruitment to inflamed tissues and their ability to transmit the virus to CD4 T cells, potentially offering new avenues for HIV prevention and treatment.

## Introduction

A key event during HIV and SIV transmission is the rapid recruitment of plasmacytoid dendritic cells (pDC) from blood to the mucosal sites of HIV/SIV exposure [1] where they produce type I interferons (IFN-I) as one of the first innate immune defences [2]. pDCs have been historically associated with antiviral responses mediated by their: i) production of IFN-I α and β and subsequent IFN stimulated gene (ISG) expression; ii) production of chemokines and cytokines that recruit the primary HIV target CD4 T cells to sites of infection [1,3]; and iii) antigen presentation and priming of T cells. Therefore, pDCs may act as a 'double-edged sword' in initial HIV infection where production of inflammatory cytokines and chemokines to recruit CD4 T cells is countered by the antiviral protection provided by IFN-I to limit viral spread.

Recent studies utilising single-cell RNA sequencing (scRNAseq) have redefined the repertoire of DCs in blood, revealing a novel myeloid DC subset termed AXL[+] Siglec-6[+] (AS) DCs. These ASDCs express classical pDC and DC markers, and can be subdivided into CD11c and CD123 expressing subsets [4]. Therefore, pDCs have now been redefined as IFN-I producing cells with minimal capacity to induce T cell activation, proliferation [4,5] and the previously reported pDC responses to HIV might be attributed to ASDCs. Indeed, CD11c[+] ASDCs have been shown to produce IL-12 p40 upon toll-like receptor (TLR)7 stimulation whilst pDCs and CD123[+] ASDCs did not [4]. As such, to explore new avenues to block HIV transmission, it is now critical to define the role that pDCs and both ASDC subsets play in initial infection. CD123[+] ASDCs in blood have been reported to be permissive to infection [6] via their expression of the HIV binding receptor CD169/Siglec-1 and they can transfer HIV to T cells. However, it is not known if ASDCs are present in the tissues where HIV transmission occurs or whether CD11c[+] ASDCs are also HIV target cells.

Here, we investigated CD11c[+] ASDCs, CD123[+] ASDCs and pDCs and showed that: i) all three cell types are present in inflamed human skin, anogenital tissues and lymph nodes; ii) they express the HIV chemokine entry receptors CCR5 and CXCR4; iii) both ASDC subsets express costimulatory molecules and stimulate T cell proliferation, activation, and polarisation more toward Th2, Th9, Th17, Th22, Treg and less toward Th1; iv) pDCs are the predominant antiviral IFN and pro-inflammatory cytokine (TNF-α and IL-6) producing cells and that they also secrete the CCR5 binding chemokine CCL3-5 which recruit CD4 T cells to sites of infection while inhibiting HIV entry by binding to CCR5 [7]; and v) both ADSC subsets transfer

HIV to CD4 T cells in two distinct phases associated with C-type lectin receptor (CLR) mediated uptake and productive infection, while pDCs transferred HIV via CLR-mediated uptake only.

## Results

### Identification of AXL⁺ Siglec-6⁺ DCs and plasmacytoid DCs in human blood

We tested the human pDC isolation kit II and a Pan DC enrichment kit to assess which kit can enrich for CD11c⁺ and CD123⁺ ASDC subsets from human peripheral blood mononuclear cells (PBMCs) [4]. Before using any kits, we detected pDCs and ASDCs in very low proportions in PBMCs as of the 96.8% Lin1⁻HLA-DR⁺CD141$^{lo}$ cells, 2% were ASDCs and 37% were pDCs (Fig 1a, **top row**).

Using the pDC isolation kit, which relies on BDCA2 selection, 98% of the Lin⁻ HLA-DR⁺ cells were pDCs, 0.9% were CD123⁺ ASDCs and no CD11c⁺ ASDCs were detected (Fig 1a, **middle row**). The presence of the CD123⁺ ASDC population was confirmed by t-stochastic neighbour embedding (t-SNE) analysis of the flow cytometry data, which showed a distinct population from the pDC cluster (S1a Fig), and heat map visualisation of the t-SNE plots confirmed the expression of AXL, Siglec-6 and CD123 but not CD11c (S1b Fig). Therefore, only CD123⁺ ASDCs could be isolated using the pDC isolation kit II.

Using the human Pan DC enrichment kit, which negatively selects for CD141⁺ cDC1, CD11c⁺ cDC2 and BDCA2⁺ pDC, we identified pDCs and both subsets of ASDCs (Fig 1a, **bottom row**). Of the Lin1⁻HLA-DR⁺ cells, 21% were pDCs and less than 2% of each ASDC subset were present (Fig 1b–1c). t-SNE analysis of the flow cytometry data verified that pDCs, CD11c⁺ and CD123⁺ ASDCs represented distinct populations of Lin1⁻HLA-DR⁺ cells. pDCs were grouped in the bottom right cluster, whilst CD123⁺ ASDCs were a discrete peninsula of the pDC cluster and CD11c⁺ ASDCs clustered together but on a separate island (Fig 1d). Heat map visualisation of the t-SNE analysis also confirmed the phenotype of each population. AXL and Siglec-6 co-expression was only identified on ASDC clusters, whilst BDCA2 and CD123 expression was mainly associated with pDCs and CD123⁺ ASDCs (Fig 1e). CD11c expression was limited to the left island containing the CD11c⁺ ASDCs, which were also clearly separated from the far-right cluster of CD141⁺ cDC1 based on CD141 expression (Fig 1e). Back-gating on each population demonstrated that CD11c⁺ ASDCs had slightly higher SSC, CD45 and HLA-DR expression compared to other cell types (Fig 1f). CD11c⁺ ASDCs also expressed lower levels of AXL, Siglec-6 and BDCA2 than CD123⁺ ASDCs, as reported by others [4]. Thus, only the Pan DC enrichment kit successfully isolated pDCs, CD11c⁺ and CD123⁺ ASDCs from PBMCs.

### Transcriptomic profiling of blood derived AXL⁺ Siglec-6⁺ DCs and plasmacytoid DCs

We used NanoString to define the ASDC and pDC genomic signatures on FACS-sorted blood-derived pDCs and both subsets of ASDCs (Fig 2a). Since our NanoString panel was targeting genes involved in immunology, we were limited in identifying all genes that define pDCs, ASDCs and all those that discriminate between CD123⁺ and CD11c⁺ ASDCs as reported by Villani et al. [4]. After normalizing the NanoString data, *TLR7* was detected in pDCs only; *IRF8*, *IRF7*, *APP*, *TCF4* and *LILRA4* were expressed more highly in pDCs and CD123⁺ ASDCs; *CD5* was exclusively identified in ASDCs. Out of all genes that discriminated between CD123⁺ and CD11c⁺ ASDCs, our NanoString panel showed *LILR4* expression only

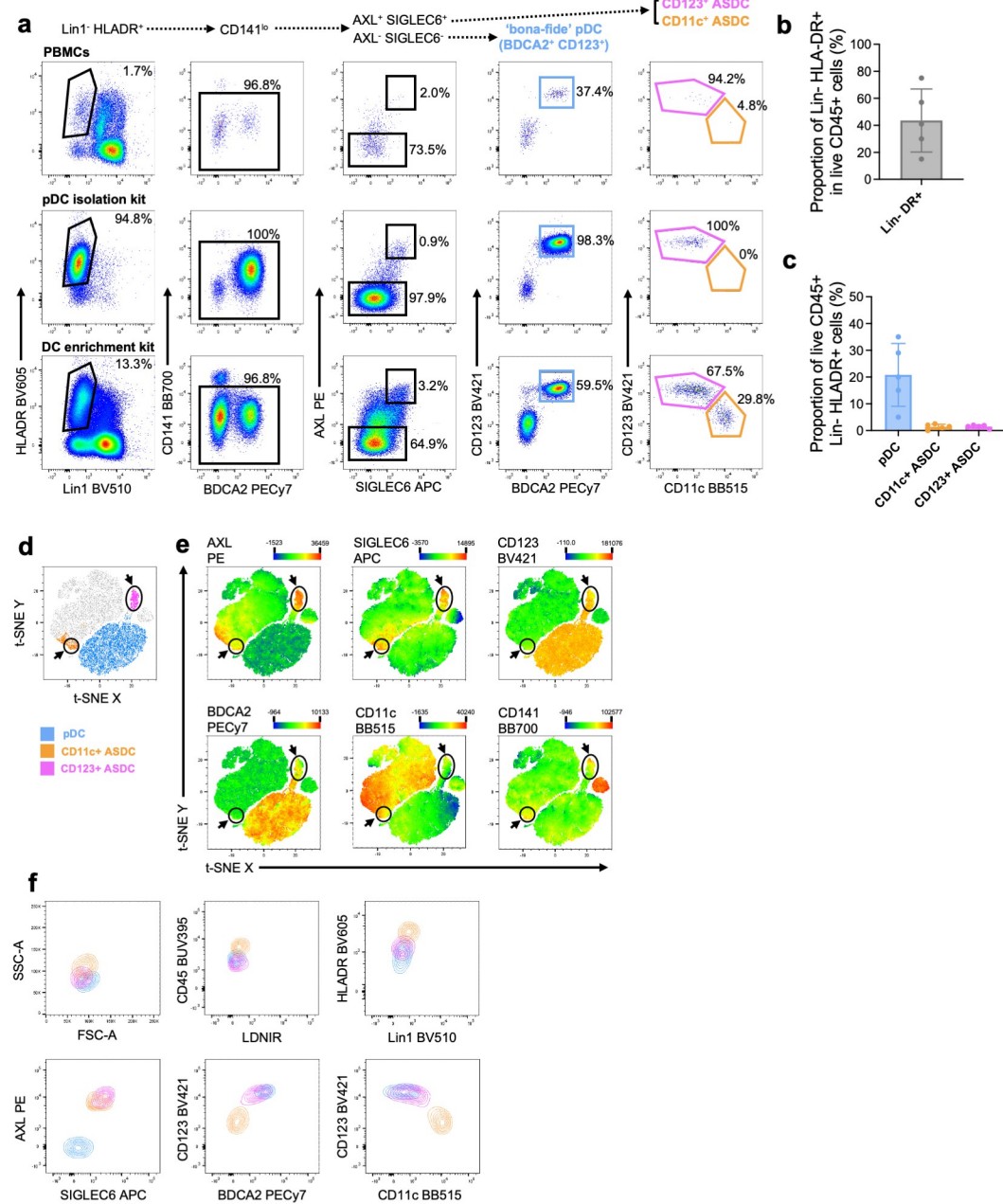

**Fig 1. Identification of pDC and ASDCs in PBMCs using pDC Isolation Kit and DC Enrichment Kit. (a)** Representative dot plots showing the gating strategy to identify pDCs (AXL$^-$ Siglec-6$^-$ BDCA2$^+$ CD123$^+$), CD123$^+$ ASDCs (AXL$^+$ Siglec-6$^+$ CD123$^+$ CD11c$^{-/lo}$) and CD11c$^+$ ASDCs (AXL$^+$ Siglec-6$^+$ CD11c$^+$ CD123$^{-/lo}$). Top row demonstrates isolation from PBMCs, middle row uses Human pDC Isolation Kit II and bottom row uses Human Pan DC Enrichment Kit. **(b)** Mean proportion of Lin1$^-$ HLA-DR$^+$ cells and **(c)** mean proportion of pDCs, CD11c$^+$ and CD123$^+$ ASDCs using the Human Pan DC enrichment Kit (±SD, n = 5). **(d)** A representative distribution of pDCs (blue), CD11c (orange) and CD123 ASDCs (purple) on a t-SNE plot to verify the phenotypes of blood pDCs and ASDCs isolated via the Human Pan DC Enrichment Kit. **(e)** Heat map visualisations of the median fluorescence intensity of surface AXL, Siglec-6, CD123, BDCA2, CD11c and CD141 expression for the populations shown on t-SNE plot. **(f)** Representative contour plots showing the phenotypic characteristics of pDCs, CD11c$^+$ and CD123$^+$ ASDCs.

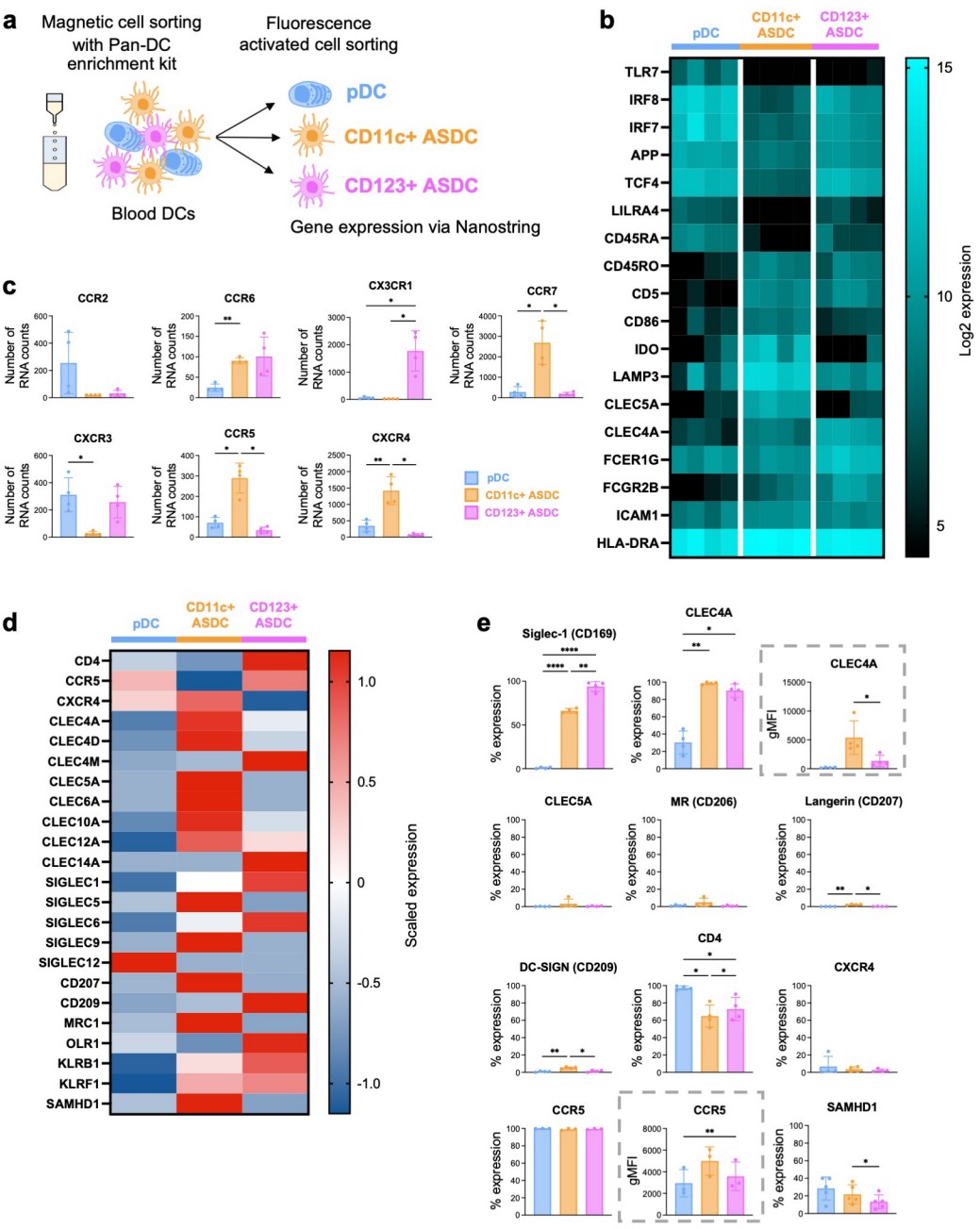

**Fig 2. Transcriptomic and proteomic profile of pDCs and ASDCs. (a)** Workflow for isolating pDCs and ASDCs using the Pan DC enrichment kit followed by FACS cell sorting to separate pDCs, CD123+ ASDCS and CD11c+ ASDCs. RNA was extracted and processed via NanoString. **(b)** Heat map for genes delineating pDCs from ASDCs (n = 4). **(c)** Chemokine gene expression by NanoString in pDCs and ASDCs. Data are presented as ±SD (n = 4). Statistical analysis was performed using one-way ANOVA with Tukey's multiple comparisons test, *p< 0.05; **p< 0.01. **(d)** Gene expression of HIV binding and uptake receptors on blood derived pDCs and ASDCs. Blood PBMCs (GSE94820) [4] were transcriptionally profiled by scRNAseq. pDC, CD11c+ ASDC and CD123+ ASDC annotations were determined using metadata provided by the authors. All genes pertaining to HIV binding lectin receptors, HIV entry receptors and SAMHD1 were compared between pDCs, CD11c+ ASDCs and CD123+ ASDCs and calculated as a scaled value. The scaled values were plotted on a heat map using Prism. **(e)** The expression of Siglec-1 (CD169), CLEC4A, CLEC5A, MR (CD206), Langerin (CD207), DC-SIGN (CD209), CD4, CXC4, CCR5 and SAMHD1 were determined on each cell type immediately after their isolation from blood. The percentage expression of each subset was calculated for each marker, and gMFI (grey dotted box) calculated when percent expression exceeded 80% for multiple subsets (n = 3–4). *p < 0.05, **p < 0.01 by one-way ANOVA with Tukey's multiple comparisons test.

in CD123[+] ASDCs. As all the above genes show a similar pattern to that reported by Villani et al [4] in differentiating ASDCs and pDCs, this implies that the gene signature for ASDCs did not change upon 18 h of culture prior to NanoString testing. We also noted pDCs were CD45RA[+], CD11c[+] ASDCs were CD45RO[+] while CD123 ASDCs were a mix of CD45RO[+] and RA[+]. A higher expression of the costimulatory molecule *CD86* was detected in CD11c[+] ASDCs compared to CD123[+] ASDCs. This also correlated with its higher protein expression on CD11c[+] ASDCs as previously reported [4]. Interestingly, we also identified *indoleamine 2,3-dioxygenase (IDO)*, *CLEC5A* and *LAMP-3* as being highly expressed by CD11c[+] ASDC, while CLEC4A, *FCER1G* and *FCGR2B* were more expressed in CD123[+] ASDCs. Furthermore, our data showed that *ICAM-1* and *HLA-DR* were expressed by pDCs and ASDCs.

In summary, we have identified some characteristic genes that identify sorted blood pDCs and the two subsets of ASDCs.

## Chemokine receptor expression that mediates AXL[+] Siglec-6[+] DCs and plasmacytoid DCs migration to peripheral tissue sites

Migration of immune cells into tissues occurs via a number of chemotactic gradients mediated by the expression of the chemokine receptors CCR1-2, CCR5-8, CXCR1-4, CXCR6 or CX3CR1 [8]. To examine which chemokine receptor(s) may mediate ASDC and pDC migration from blood to inflamed tissues, we interrogated our NanoString data to assess the expression of these receptors (Fig 2c). The following gene expression profiles were noted: i) *CCR2* by pDCs only; ii) *CCR6* by both ASDC subsets; iii) *CX3CR1* by CD123[+] ASDCs only; iv) *CCR7* by CD11c[+] ASDCs; v) *CXCR3* by pDCs and CD123[+] ASDCs; vi) *CCR5* and *CXCR4* more highly expressed in CD11c[+] ASDCs followed by pDCs and CD123[+] ASDCs; vii) *CCR8* was not detected in any subset. As our NanoString assay did not include *CCR1, CCR3, CCR4, CXCR1-2*, we analysed the scRNAseq data of blood derived ASDCs from Villani et al [4]. We found that *CCR1* and *CXCR1-2* were expressed only by pDCs while *CCR3-4* were detected in CD123[+] ASDCs only (S2 Fig). We also confirmed that the chemokine gene expression profiles identified by our NanoString data were similar to this RNAseq dataset. In summary, pDCs and both subsets of ASDCs expressed the genes of many chemokine receptors to allow their migration to inflamed peripheral tissue sites.

## Blood-derived AXL[+] Siglec-6[+] DCs and plasmacytoid DC expression of HIV binding and entry receptors

To understand HIV interactions with ASDCs and bona-fide pDCs (depleted of CD123[+] ASDCs), we examined the gene and surface expression of: i) lectin binding receptors that mediate endocytic uptake including those known to bind HIV: CD169/Siglec-1, CD209/ DC-SIGN, CD206/MR and CD207/langerin; ii) the HIV entry receptors CD4, CCR5, and CXCR4 leading to productive infection; and iii) the myeloid cell retroviral restriction nuclear dNTPase SAMHD1. For lectin expression, our NanoString data was limited to *CLEC4A* and *CLEC5A*. Therefore, we again interrogated the publicly available scRNAseq data of blood derived ASDCs produced by Villani et al [4] (Fig 2d) to investigate the full CLR repertoire. Except for *Siglec-12*, we found that pDCs expressed very low levels of the genes encoding almost all lectin receptors, CD11c[+] ASDCs expressed the highest levels of most CLRs, notably the known HIV binding CLRs *CD207/langerin*, CD206/*MR* as well as *CLEC4A, CLEC5A* and *CLEC10A*. We also noted that CLRs expressed by CD11c[+] ASDCs were expressed at much lower levels by CD123[+] ASDCs, however the latter expressed other CLRs at high levels including the HIV binding lectins *CD169/Siglec-1* and *CD209/DC-SIGN*. The gene expression of

HIV entry receptors *(CD4, CCR5 and CXCR4)* was detected at various levels in all three subsets. Finally, *SAMHD1* was more highly expressed in CD11c[+] than CD123[+] ASDCs and pDCs.

We then used high parameter flow cytometry to measure surface protein expression levels. In correlation with our gene expression analysis, pDCs did to not express any of the CLRs we measured except CLEC4A. CD169/Siglec-1 was highly expressed by CD123[+]ASDCs. CD11c[+] ASDCs expressed most CLRs including CLEC4A at high levels, while CLEC5A, CD206/MR and CD207/langerin were expressed at low levels (Figs 2e and S3). Furthermore, despite higher gene expression of DC-CD209/SIGN by CD123[+] ASDCs (Fig 2d), CD11c[+] ASDC expressed more DC-SIGN on their surface (Figs 2e and S3). For HIV entry receptors, pDCs expressed the highest levels of CD4, whereas CXCR4 was expressed at low levels (in contrast to its gene expression) and CCR5 at high levels by all three cell subsets, though CCR5 fluorescent intensity (gMFI) was highest on CD11c ASDCs (Fig 2e). Finally, SAMHD1 was expressed most highly by CD11c[+] ASDCs (Fig 2e). In summary, CD11c[+] ASDCs expressed the highest levels of CD206/MR, CD207/langerin and CD209/DC-SIGN. All subsets expressed the primary HIV entry receptor, CD4, and chemokine co-receptors CCR5 and CXCR4.

## Identification of AXL[+] Siglec-6[+] DCs and plasmacytoid DCs in inflamed human tissues

To date, ASDCs have been described in PBMCs and tonsillar tissue [4], spleen [5], inflamed cerebrospinal fluid [9], inflamed broncho-alveolar lavage [10] and inflamed skin [11]. To determine if they are present in tissues affected by other inflammatory diseases, we used our blood ASDC-specific gene of Fig 2b to cross reference several publicly available human tissue scRNAseq datasets of cells derived from ulcerative colitis [12,13], atopic dermatitis [14], psoriasis [15], colorectal cancer [16] and vagina [17]. The sub-setting steps of the scRNAseq analysis are shown in S4 Fig. We detected two pDC and two ASDC clusters in psoriasis affected skin (Fig 3a–3b) and colorectal cancer (Fig 3c–3d). One pDC cluster appeared to be more mature (*CD83*[+]), with decreased expression of *IRF8* and *IRF7*. CD11c[+] ASDCs expressed *IDO1*, and *LAMP3*, while the CD123[+] ASDCs expressed *CLEC4A*, *FCER1G* and *FCGR2B*. We found equal proportions of CD123[+] and CD11c[+] ASDCs in psoriasis (Fig 3a) while in colorectal cancer there were more CD123[+] than CD11c[+] ASDCs (Fig 3c). In colorectal cancer, there were more immature than mature pDCs and vice versa for psoriasis. Immune cell enrichment was performed on both the psoriasis and colorectal cancer data sets, but not the others. As ASDCs are present in very low numbers, this lack of enrichment may account for our inability to identify ASDCs in ulcerative colitis, atopic dermatitis, and vagina via RNAseq. As such, further studies are required to confirm the presence of ASDCs in these tissues and disease settings.

With ASDCs presence in inflamed tissue confirmed via transcriptomics, we set out to determine if ASDCs were also present in fresh human tissue by flow cytometry. Initially, we obtained inflamed rectal tissue from patients undergoing surgery for a range of inflammatory intestinal conditions (S2 Table). Immune cells were isolated using our optimised enzymatic digestion protocols for intestinal tissues [18] where samples are processed within 15 minutes of their removal from the body. ASDCs and pDCs were identified using high parameter flow cytometry an inflamed rectums affected by active ulcerative colitis. Similar to blood, tissue pDCs were in greater proportion than ASDCs (Fig 4a–4c). Back gating was performed to show that pDCs, and both CD11c[+] and CD123[+] ASDCs were present (Fig 4b). We also detected pDCs and ASDCs in inflamed mesenteric lymph nodes (Fig 4d), but neither cell type was present in non-inflamed rectum (Fig 4e). This was confirmed *in situ* in one inflamed colon affected by diverticulitis using imaging mass cytometry where pDCs and both ASDC subsets were detected and were mostly present in submucosal lymphoid follicles (Fig 4f). Furthermore, we identified both

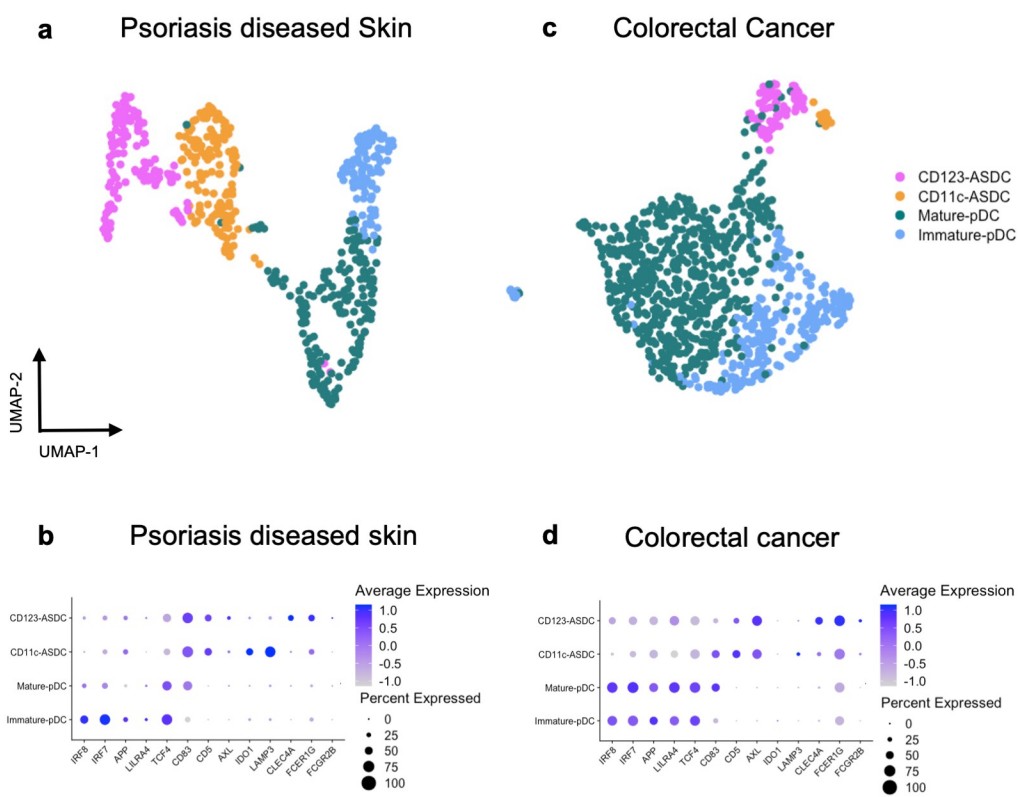

**Fig 3. Transcriptomic profile of pDCs and ASDCs via scRNASeq.** UMAP plots and expression of genes delineating pDCs from ASDCs in data sets of cells derived from inflamed tissues of psoriasis diseased skin (E-MTAB-8142[15]) is shown in **(a-b)** and from colorectal cancer (GSE178341[16]) in **(c-d)**.

ASDC subsets in very low numbers in the epidermis and underlying dermis of inflamed human tissues including abdominal skin, labia, and outer foreskin and in the epithelium and lamina propria of inner foreskin (S3 Table, Fig 4g). cDC2s were additionally identified in these samples as a separate population to ASDCs and in much higher numbers. Functional assays on tissues derived ASDCs were impossible due to the low number of isolated cells. In summary, we have shown that ASDCs are present in inflamed human anogenital tissues.

## Expression of HIV binding and entry receptors on AXL⁺ Siglec-6⁺ DCs and plasmacytoid DCs isolated from inflamed human tissues

CLR surface expression was assessed by flow cytometry on the pDCs and ASDCs isolated from epidermis and dermis/lamina propria of inflamed human abdominal skin, labia, and inner/outer foreskin (Fig 5a). Higher expression of CD169/Siglec-1 was detected on CD123⁺ rather than CD11c⁺ ASDCs while pDCs had very low to no expression. CD209/DC-SIGN, CD206/MR and CD207/langerin were expressed more highly on CD11c⁺ than CD123⁺ ASDCs and pDCs. CD4 was expressed at high levels in all three cell subsets. This preliminary data indicates that blood (Fig 5b) and tissue ASDCs express similar levels of Siglec-1 and CD4. However, expression of the CLRs CD206/MR, CD207/Langerin and CD209/DC-sign trended higher on tissue ASDCs compared to blood. We have also identified cDC2 in tissues (Fig 4g) and blood (S5 Fig). In tissues, cDC2 differed in CLR expression compared to CD11c⁺ ASDCs (Fig 5) as they had low Siglec-1 and no DC-sign expression.

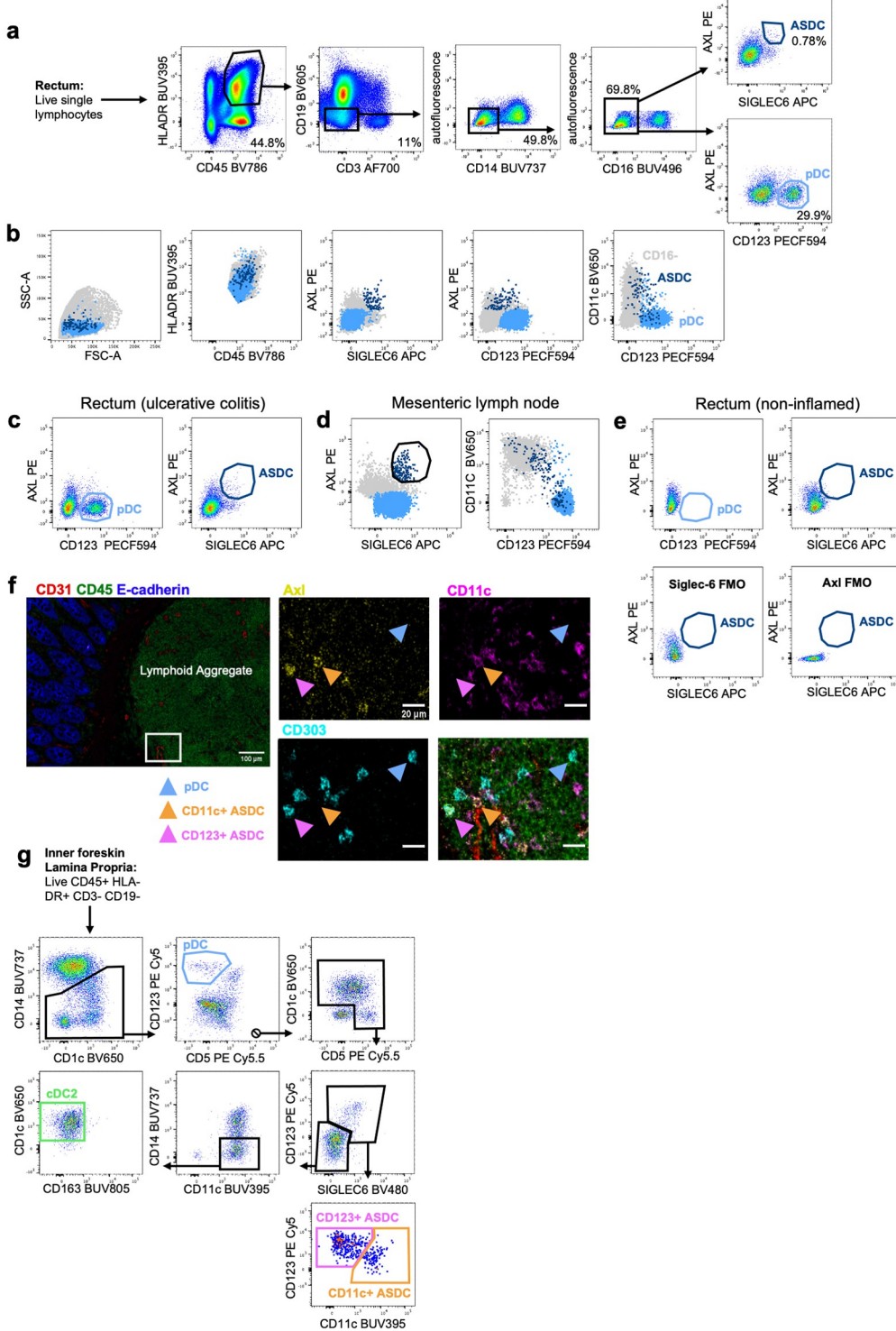

**Fig 4. Identification of pDCs and ASDCs in human tissues. (a)** Representative dot plots identifying pDCs (AXL$^-$ CD123$^+$) and ASDCs (AXL$^+$ Siglec-6$^+$) in a rectum of a patient with ulcerative colitis. **(b)** Back gating to show the surface expression of pDCs and ASDC and to confirm the phenotype detected in a. **(c-d)** Representative data showing pDCs and ASDCs in the rectum of patient with chronic ulcerative colitis in c and in mesenteric lymph nodes of inflamed rectum in d. All cells were first gated as live HLA-DR$^+$ CD45$^+$ CD19$^-$ CD3$^-$ Autofluorescent$^-$ CD14$^-$ CD16$^-$ cells. **(e)** Absence of pDCs and ASDCs in non-inflamed rectal tissue (top) after adopting the gating strategy listed in a.

Siglec-6 FMO and AXL FMO (bottom) confirmed real expression. (f) ASDCs AXL⁺ CD303⁺ (pink arrows), ASDCs AXL⁺ CD11c⁺ (orange arrows) and pDCs (blue arrows) were identified by microscopy at the periphery of a submucosal lymphoid aggregate in an inflamed human colon. ASDCs were located proximal to a blood vessel based on CD31 expression. (g) Representative gating strategy as demonstrated by inner foreskin lamina propria to identify pDCs, ASDCs and cDC2s. All cells were first gated as live CD45⁺ HLA-DR⁺ CD3⁻ CD19⁻; pDCs (blue) were defined as CD14ˡᵒ CD5⁻ CD123⁺, CD123⁺ ASDCs (pink) as CD14ˡᵒ CD5⁺ CD1c⁺ Siglec-6⁺ CD123⁺, CD11c⁺ ASDC (orange) as CD14ˡᵒ CD5⁺ CD1c⁺ Siglec-6⁺ CD11c⁺ and cDC2 (green) as CD14⁻ CD5⁺ CD1c⁺ CD11c⁺ CD163⁻. This gating strategy differed to the gating in a-e as a different panel design was used to allow for identification of cDC2s.

## Functional phenotyping and polarisation of T cells by blood-derived AXL⁺ Siglec-6⁺ DCs and plasmacytoid DCs

DCs are highly efficient antigen presenting cells (APC) and deliver antigens to activate naïve T cells driving them to be polarised into various CD4 T helper (Th) subsets or effector CD8 T cells. Therefore, we next assessed the surface expression of costimulatory markers on blood derived pDCs and ASDCs that are required for DCs to perform these functions. pDCs did not express CD80, CD83 or CD86 on their surface. Compared to ASDCs, they expressed low levels of HLA-DR and the T cell adhesion marker CD54/ICAM-1 (Fig 6a). Similar to blood cDC1 and cDC2 [19], both ASDC subsets did not express CD80 or CD83. However, 80% of both subsets expressed CD86 and CD54/ICAM-1 with CD11c⁺ ASDCs expressing higher levels of HLA-DR than CD123⁺ ASDCs. It is worth noting that low mRNA levels can negatively

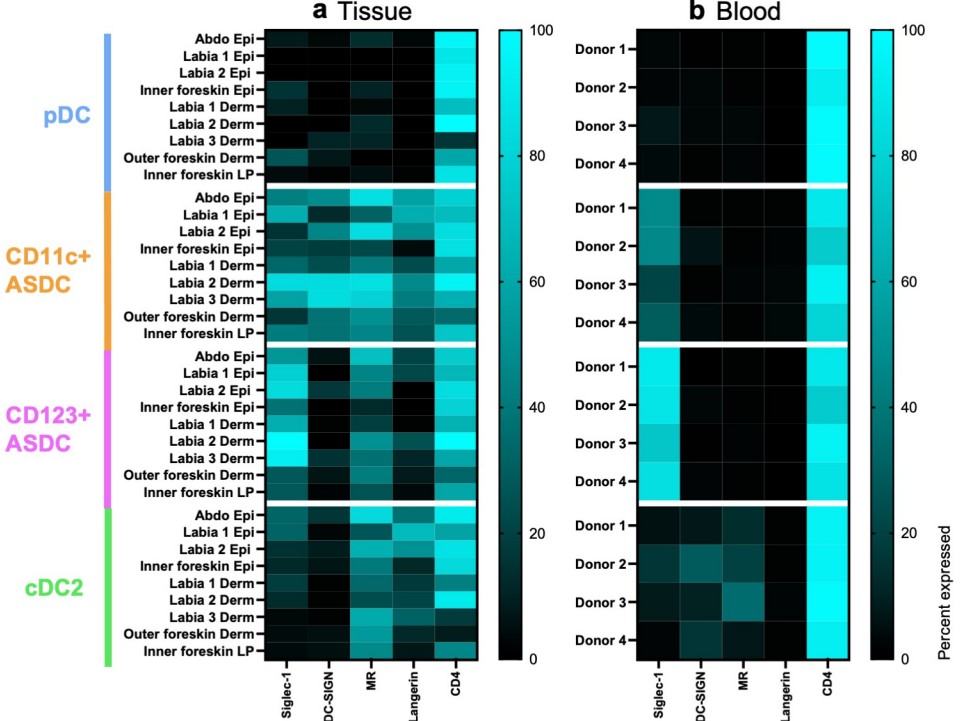

Fig 5. Expression of key HIV binding receptors on ASDCs, pDCs and cDC2s from human tissues and blood. (a) Immune cells liberated from abdominal epidermis (abdo epi) (n = 1), labia epidermis (n = 2), inner foreskin epidermis (n = 1), labia dermis (derm) (n = 3), outer foreskin dermis (n = 1) and inner foreskin lamina propria (LP) (n = 1) were stained for flow cytometry. The percentage expression of Siglec-1, DC-SIGN, MR, Langerin, and CD4 on pDCs, CD11c⁺ ASDCs, CD123⁺ ASDCs and cDC2s for each tissue was plotted on a heat map and compared to blood-derived pDCs, ASDCs and cDC2s in (b).

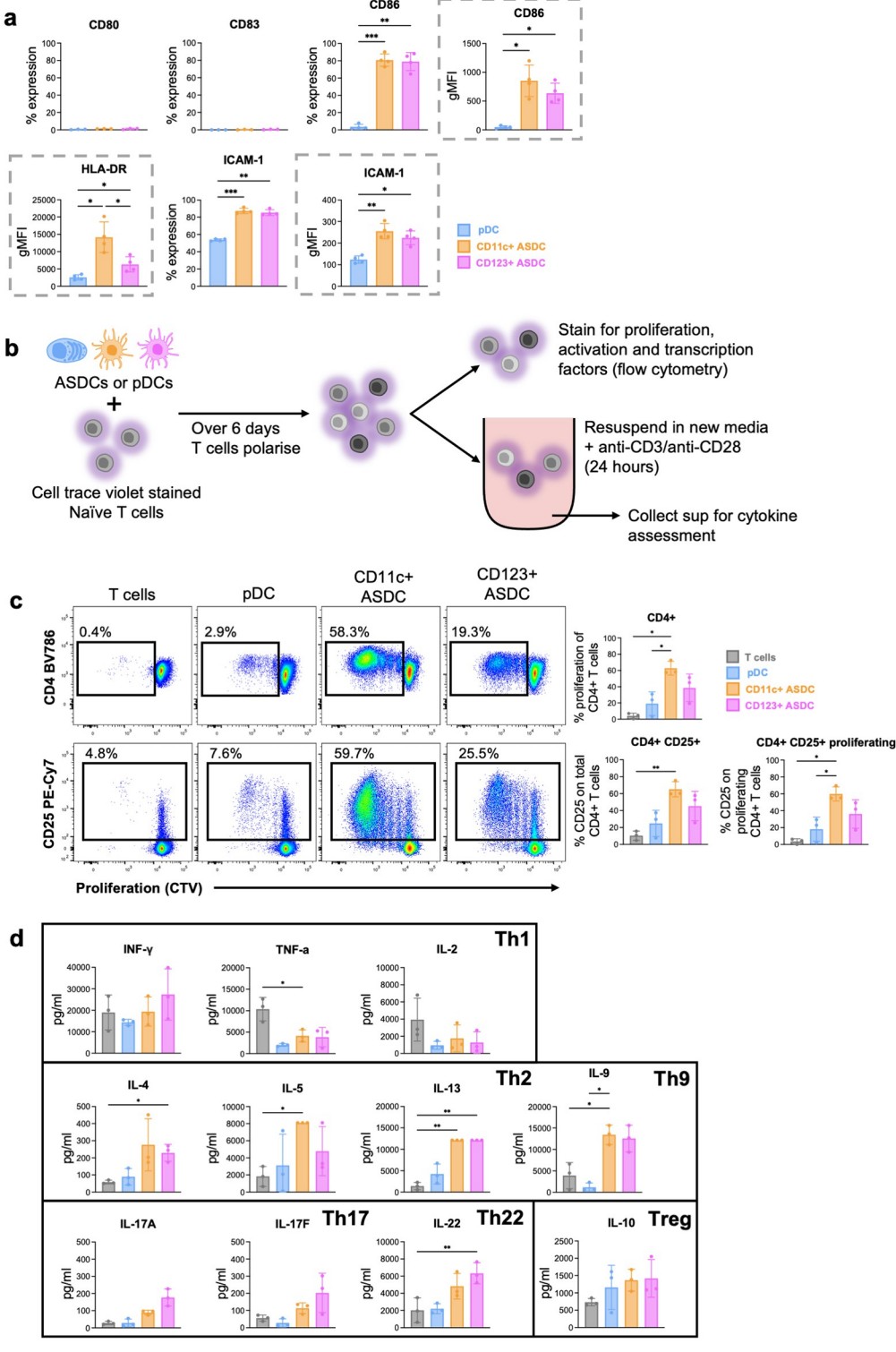

**Fig 6. Functional phenotyping of blood-derived pDCs and ASDCs. (a)** Cell surface expression of CD80, CD83, CD86, HLA-DRM and ICAM-1 was determined on each cell type immediately after their isolation from blood. The percentage of expression and gMFI (when % of expression was above 80%, outlined in grey dotted box) of each subset was calculated. **(b)** Workflow for functional phenotyping. **(c-d)** FACS sorted pDCs, CD123 and CD11c ASDC were cultured at 37˚C with Cell trace Violet-stained naïve T cells at a ratio of 1 ASDC or pDC: 10 T cells. On day 6, cultures were analysed by flow cytometry to assess CD4 T cell **(c)** proliferation via CTV stain and activation via CD25

expression. (**d**) on day 6, all conditions were treated with anti-CD3/CD28 for 24 hours. Supernatants were collected to assess cytokine production. Data is presented as ±SD (n = 3–4). Statistical analysis was performed using one-way ANOVA with Tukey's multiple comparisons test. *p < 0.05; **p < 0.01; ***p < 0.001.

correlate to high level of protein expression and vice versa. This was the case for CD83 and CD86. CD83 was expressed by all cells at the RNA level in S1 Table but was not detected at the cell surface in Fig 6a. However, CD86 which had lower mRNA counts than CD83 in all subsets was expressed at the cell surface as per the RNA counts i.e. higher gMFI on CD11c$^+$ ASDCs, followed by CD123$^+$ ASDCs, then pDCs.

We next assessed the capacity of pDCs and ASDC to stimulate allogeneic naïve T cell proliferation, activation, and Th polarisation (Fig 6b). Corresponding with their surface expression profiles, pDCs were very poor inducers of CD4 (Fig 6c) and CD8 T cell activation and proliferation (S6a and S6b Fig), whereas both ASDC subsets could perform these functions, with CD11c$^+$ ASDCs inducing more CD4 and CD8 T cell activation and proliferation than CD123$^+$ ASDCs, aligning with a previous report by Alcantara-Hernandez et al [20]. We also examined T cell cytokine production and possible subset polarisation by pDCs and ASDCs (Fig 6d). pDCs induced low levels of all Th cell cytokines. Compared to T cells that were not polarised with any APCs, ASDCs stimulated low levels of Th1 (TNF-α, IL-2) cytokines and higher levels of Th2 (IL-4, IL-5 and IL-13), Th9 (IL-9) and Th22 (IL-22) cytokines. IL-17 secretion from Th17 and IL-10 from Treg was low but higher than non-polarised T cells. IFN-γ secretion was high but similar to non-polarised T cells. Selected expression of transcription factors Tbet, FoxP3, RORγt, which identify Th1, Treg, and Th17, respectively, was examined. ASDCs induced the highest proportion of T cells expressing RORγt followed by FoxP3 and low T bet expression (S6c Fig).

In summary, both subsets of ASDCs, expressed the co-stimulatory molecules CD86, ICAM-1 and HLA-DR at higher levels than pDCs, with CD11c$^+$ having the highest expression. pDCs stimulated very low levels of naïve T cell allo-proliferation whereas ASDCs induced a much higher level of proliferation. Both ASDC subsets polarised naïve T cells more toward Th2, Th9, Th22, Th17 and Treg but less toward Th1.

## Chemokine and cytokine production in blood-derived AXL$^+$ Siglec-6$^+$ DCs and plasmacytoid DCs in response to HIV

As anogenital inflammation is now thought to be a pre-requisite to HIV transmission [21], and ASDCs are only found in inflamed tissues, we next investigated ASDCs in the context of HIV infection. To this end, we either mock or HIV treated pDCs and ASDCs for 18 hours to define any HIV induced transcriptional changes using NanoString. Firstly, we compared HIV to mock treated cells and found that all characteristic genes for blood pDCs and ASDCs identified in the heat map of Fig 2b were downregulated upon HIV exposure, except for *CD86* and *ICAM-1*, which were upregulated in HIV treated CD11c$^+$ and CD123$^+$ ASDCs, respectively (S7a Fig).

We then investigated the expression of chemokines and cytokines and measured their secretion into the culture supernatant using a LEGEND plex bead array. In response to HIV treatment, pDCs increased their gene expression and secreted more of the CCR5 binding chemokines CCL3 (MIP1α), CCL4 (MIP1β), CCL5 (RANTES) and CXCL10 compared to ASDCs. CXCL8/IL-8 known to be produced in response to viruses was expressed and secreted at significantly greater levels by CD11c$^+$ ASDCs when compared to pDCs and CD123$^+$ ASDCs (Figs 7 and S7b). The pro-inflammatory cytokines TNF-α and IL-6 were secreted at high levels

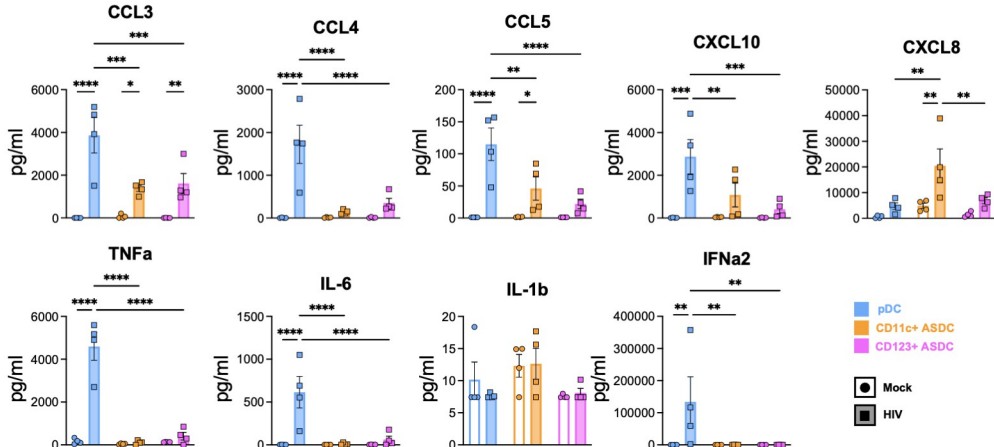

**Fig 7. Chemokine and cytokine production after HIV interaction with pDCs and ASDCs.** LEGENDplex assay was carried out to determine the concentration of cytokines in the supernatants of mock and HIV-1 BaL treated pDCs and ASDCs at 18 hours post HIV treatment. Data presented as ±SD (n = 4). Statistical analysis was performed using one-way ANOVA with Tukey's multiple comparisons test. *p< 0.05; **p< 0.01; ***p< 0.001; ****p< 0.0001.

by HIV exposed pDCs but not from ASDCs, even though ASDCs expressed the TNF-α gene (S7c Fig). The proinflammatory cytokine IL-1β was detected at very low levels in mock and HIV treated pDCs and ASDCs at the protein level, and only in mock CD11c⁺ ASDCs at gene level (S7c and S7 Fig). There was no gene or protein detection of IL-10, IL-12A, IL-13, IL-17, IL-18, or IL-33. pDCs expressed *IFN-α*, *β* and *λ1* in response to HIV (S7d Fig) and IFN-α2 was secreted at high levels. However, ASDCs did not produce any interferons in response to HIV (Fig 7). Finally, IFN-stimulated genes (ISG) including *IFTM1*, *IFIT2*, *MX1* and *IFI35* were upregulated in HIV exposed pDCs and both ASDC subsets (S7e Fig), despite the lack of IFN production in the latter. This was similar to HIV induction of ISGs in DCs and macrophages in the absence of IFN as we have previously reported [22,23].

In summary, pDCs were the main producers of CCR5 binding chemokines, interferons and proinflammatory cytokines in response to HIV, while CD11c⁺ ASDCs produced only high levels of CXCL8.

## HIV viral transfer from blood-derived AXL⁺ Siglec-6⁺ DCs and plasmacytoid DCs to CD4 T cells

We have shown in several key studies that human tissue DCs and Langerhans cells function to capture HIV and then transfer the virus to CD4 T cells in two phases, an early first phase which is dependent on CLR-mediated endocytic uptake (and not infection) followed by a later second phase which is dependent on CD4/CCR5 mediated entry and infection. As ASDCs are present in extremely low numbers in human tissue, it was impossible to carry out uptake and infection assays using tissue derived ASDCs. Therefore, we next assessed the ability of blood-derived pDCs and each ASDC subset to mediate first phase (2 hours post HIV exposure) and second phase (96 hours post HIV exposure) transfer of HIV to CD4 T cells.

We observed two successive phases of HIV viral transfer to T cells from ASDCs. Both pDCs and ASDCs mediated first phase transfer but interestingly, CD11c⁺ ASDCs showed a trend of higher capacity for binding HIV (Fig 8a). Contrastingly in second phase, CD123⁺ ASDCs showed a trend of more virus transfer to T cells than CD11c⁺ ASDC while pDCs did not, suggesting that CD123⁺ ASDCs support the highest levels of productive infection. This was

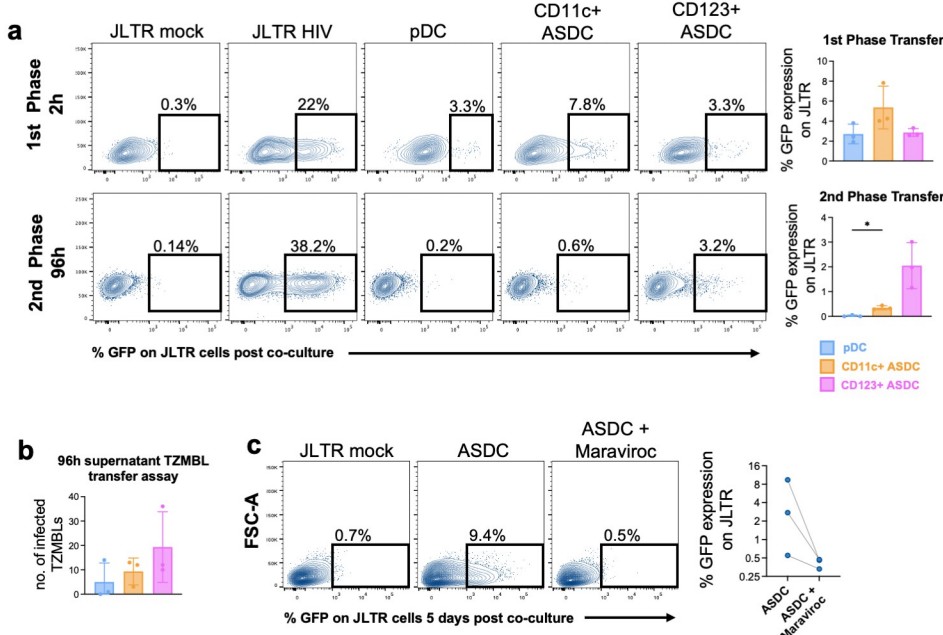

**Fig 8. HIV interactions with blood-derived pDCs and ASDCs and viral transfer to T cells. (a)** HIV transfer to JLTR cells was determined by measuring JLTR GFP fluorescence intensity by flow cytometry 2h and 96h post co-culture for first phase and second phase transfer respectively. Representative flow data and graph (data presented as ±SD, n = 3) show transfer of HIV from HIV-treated cells to JLTR (GFP⁺ JLTR). **(b)** Number of infected TZMBL cells after inoculation with supernatants collected at 96 hpi of pDCs, CD123⁺ and CD11c⁺ ASDCs. Data presented as ±SD (n = 3). **(c)** HIV transfer to JLTR cells in the presence and absence of Maraviroc. Data presented as ±SD (n = 3). Statistical analysis was performed using Wilcoxon Rank Sum Test.

confirmed by inoculating the CD4 T cell line TZMBL1 (which express *lacZ* under control of the HIV-1 promoter) with supernatants collected from infected cells after 96 hours post infection (hpi) (Fig 8b). No TZMBL infection occurred when inoculated with supernatant derived from HIV-exposed pDCs. Low TZMBL infection was detected from CD11c⁺ ASDCs supernatants but as expected, CD123⁺ ASDC supernatants gave rise to the highest levels of infected TZMBL cells, although this did not reach a statistical difference. Furthermore, to confirm productive infection of ASDCs via CD4/CCR5, we demonstrated that the CCR5 inhibitor maraviroc was able to completely block the ability of ASDCs to mediate second phase transfer (Fig 8c).

## Discussion

### Definition of ASDCs in human blood and tissue

The CD123⁺ and CD11c⁺ ASDC populations described in this manuscript correspond to DC5 identified by Villani et al.[4] which express AXL, Siglec-6, CD2, CX3CR1, CD33, and CD5, similar to the phenotype described for pre-DC1 and pre-DC2 (CD33⁺ CD45RA⁺ CD123⁺ CX3CR1⁺ CD2⁺ CD5⁺ Siglec-6⁺) identified by See et al. [5]. Based on our data of Fig 1f showing CD11c⁺ ASDCs express lower levels of AXL and Siglec-6 than CD123⁺ ASDCs and as reported by Villani et al., we believe that pre-DC1 are the CD123⁺/high CD11c⁻ ASDCs while pre-DC2 are the CD123⁻/lo CD11c⁺ ASDCs. Therefore, ASDCs investigated in this study are the equivalent of the previously described pre-DCs and ASDCs. Furthermore, our findings

show that ASDCs are a different population to cDC2s and that they are mature in phenotype like cDC2s.

We first confirmed the presence of pDCs, CD11c[+] and CD123[+] ASDCs by flow cytometry (Fig 1) and transcriptomic profiling (Fig 2) in human blood. We then showed that these cell populations were present in a range of inflamed human cutaneous and mucosal tissues including lymph nodes, colorectal mucosa affected by inflammatory bowel disease and cancer, genital and trunk skin affected by psoriasis (Figs 3 and 4). We did not observe pDCs or ASDCs in non-inflamed tissues. Importantly, we showed that CD11c[+] ASDCs are a distinct population from bona fide CD1c[+] cDC2s in tissues (Fig 4g) and blood (S5 Fig), and they are detected at a much lower frequency than cDC2s (S3 Table). Thus, we propose that like pDCs, ASDCs infiltrate peripheral tissues during inflammation while cDC2s are present in both healthy [24] and inflamed tissues [15].

Our preliminary investigation into the cell surface expression of HIV binding receptors revealed that both tissue and blood derived ASDCs have similar high expression of CD4 and CD169/Siglec-1. However, tissue ASDCs expressed more CD209/DC-SIGN, CD206/MR and CD207/Langerin compared to blood ASDCs (Fig 5). It is well known that blood and tissue DC expression profiles can differ extensively, for instance CD206/MR is highly expressed by most tissue DCs and macrophages, but is variable in blood [25]. Furthermore, tissue derived cDC2s expressed more MR compared to their blood counterparts. They also did not express DC-SIGN and had low Siglec-1 compared to tissue derived CD11c[+] ASDCs (Fig 5). Unfortunately, the very low cell numbers of pDCs and ASDCs in inflamed human tissue explants restricted us from performing any functional assays.

## Functional profiling of ASDC subsets and plasmacytoid DCs

Our data showed that CD11c[+] ASDCs stimulated naïve T cell proliferation and activation more strongly than CD123[+] ASDCs, while a marginal stimulation of T cells by pDCs was observed (Fig 6). This is due to CD11c[+] ASDCs expressing more CD86, CD54/ICAM-1 and HLA-DR than CD123[+] ASDCs (Fig 6a), while pDCs did not express these markers or expressed them at very low levels. Our findings that ASDCs, and not pDCs, are potent inducers of T cell proliferation and activation, and that ASDCs are mature in phenotype similar to cDC2s, are in agreement with See et al. [5] and Villani et al. [4]. However, the former study did not separate the two ASDC subsets and the latter did not also detect a significant difference in bulk T cell proliferation and activation between the two ASDC subsets using TLR agonists for activation. In addition, we showed here that ASDCs polarise naïve T cells more towards Th2, Th9, Th17, Th22, Treg, and less toward Th1. As T-bet regulates the differentiation of Th1 cells and represses Th2 lineage commitment [26,27], the low level of T bet expression supports a reduced polarising toward Th1. Our high proportion of T cells expressing RORγt which does not correlate with the relatively low levels of IL-17 production is similar to Segura et al when naïve T cells were co-cultured with inflammatory DCs [28]. This may be explained by Th17 cells evolving into Th1, Treg or they can co-express either RORγT/Tbet or RORγT/FoxP3 [29] to reduce IL-17 secretion. Furthermore, the pattern of polarisation may be also linked to ASDCs high expression of CD5 (Figs 2b and S7a), which has been reported to be required for cDC2s to stimulate high naïve T-cell proliferation and preferential priming of Th2, Th17, Th22 and Treg cells while monocyte-like cDCs, which express lower CD5, polarise mainly Th1 cells [30]. Here we showed that HIV infection downregulated *CD5* (S7a Fig) and this may favor the induction of the anti-viral Th1 subset. We attempted to test this hypothesis several times, but the low ASDC numbers we could isolate made the experiment impossible to perform.

### Cytokines production in HIV exposed ASDCs and plasmacytoid DCs

Upon HIV infection, the pro-inflammatory chemokines (CCL3-5 and CXCL10) were mainly produced by pDCs followed by ASDCs except CXCL8, which was highly produced from CD11c[+] ASDCs. Similar to our data, CCL3, CXCL10 and CXCL8 were also detected by Brouiller et al [31] upon infection of AXL[+] ASDCs by NL-AD8. The pro-inflammatory cytokines (TNF-α and IL-6) and IFN were only produced by pDCs (Fig 7). Siglec-1 has been shown to inhibit IFN production via degradation of TBK1 [32]. As ASDC subsets expressed variable level of Siglec-1 (Fig 2e), this may explain the lack of IFN production by HIV exposed ASDCs due to their Siglec-1 expression (Fig 2e). While pDCs attract CD4 T cells to the site of infection via their pro-inflammatory chemokines (CCL3-5) [1, 3], they also inhibit HIV entry as these chemokines bind to CCR5 [7] and with their IFN production, they inhibit viral spread. CXCL10 induces latency [33] by dephosphorylating cofilin to promote HIV integration [34] while CXCL8 can stimulate HIV replication in CD4 T cells [35]. Based on the cytokine and chemokine profiles upon HIV infection and our data that the direct activation of CD4 T cells is attributed to ASDCs and not pDCs, strategies to inhibit ASDC recruitment and functions in inflamed mucosa should be considered.

### ASDC and HIV Transmission

pDCs have been previously shown to be productively infected by HIV [36,37], but these studies were carried out prior to the discovery of ADSCs and were therefore examining a combination of pDCs and ASDCs. Our data showed in first phase transfer that CD11c[+] ASDCs had a trend of transferring more HIV to CD4 T cells than CD123[+] ASDCs and pDCs. Importantly, we showed that this corresponded with CD11c[+] ASDCs higher expression of CLRs (Figs 2e and S3), which we have previously shown to bind HIV and mediate its uptake via CD206/MR [38], CD207/langerin [21] and CD209/DC-SIGN [39]. We then showed that CD123[+] ASDCs had a trend of transferring more HIV to CD4 T cells in the second phase (Fig 8). We attempted to directly assess the productive infection of CD123[+] ASDCs and CD11c[+] ASDCs via p24 expression. However, due to low numbers of blood derived ASDCs, we had insufficient cells for accurate measurement. The higher Siglec-1 expression on CD123[+] ASDCs compared to their CD11c[+] counterparts confirmed the role of Siglec-1 in binding HIV to facilitate productive infection as shown previously for DCs [40], macrophages [41] and ASDCs [6]. Furthermore, we found that CD11c[+] ASDCs expressed higher levels of the HIV restriction factor SAMHD1 than CD123[+] ASDCs, explaining their reduced capacity in mediating second phase transfer. Additionally, CD11c[+] ASDCs had a higher gMFI for CD86 and HLA-DR than CD123[+] ASDCs, indicating a more mature phenotype. It is known that mature DCs bind more HIV than immature cells [42] and that they are less susceptible to productive infection [6]. This also explains why CD11c[+] ASDCs are efficient at mediating first phase and not second phase transfer and perhaps an increase in the maturation of the CD123[+] ASDC subset would increase their rate of first phase HIV transfer as previously reported [6]. Finally, pDCs transferred HIV to T cells at low efficiency in first phase only. This could be explained by their expression of CLEC4A, known to bind HIV [43] for transmission in first phase, while their IFN production acts to inhibit viral replication and spread [2,44] in second phase transfer.

Ruffin et al [6] investigated the interactions of blood derived pre-DCs with HIV but their study encompassed the combined CD123[+/high] CD11c[−] and CD123[−/lo] CD11c[+/high] ASDCs as their pre-DCs were defined as HLA-DR[+] Lin[−]CD33[int] CD45RA[+] CD123[+] AXL[+]. They showed that the combined ASDC subsets took HIV via CD169/Siglec-1, become infected and transfer HIV to CD4 T cells, similar to our data of HIV infection of CD123[+/high] CD11c[−] ASDCs. Brouiller et al also investigated the interactions of blood derived ASDCs with HIV [31] either

as combined ASDC subsets (when sorted on AXL$^+$ CD123$^+$), or separated subsets (when sorted on CD11c$^+$ and CD11c$^-$). The high p24 detected in their CD11c$^+$ AXL$^+$ DCs was due to infection by HIV-1 (NL-AD8) in the presence of Vpx, thus neutralising SAMHD1. This correlates with our reasoning of high SAMHD1 expression in CD11c$^+$ ASDCs which reduces their productive infection compared to CD123$^+$ ASDCs expressing lower SAMHD1.

## Concluding remarks

In summary, we have shown for the first time that inflammatory ASDCs are present in inflamed tissues where HIV transmission occurs. This is extremely important as the body of evidence that anogenital inflammation is a causative factor in HIV transmission is now undeniable, especially in sub-Saharan Africa [45,46]. Although a dysregulated vaginal microbiome is clearly a contributing factor [47,48], the key inflammatory HIV target cells have not yet been identified. Our findings here that both ASDC subsets can mediate HIV transmission to CD4 T cells has important implications for better PrEP design. For example, current PrEP drugs that block the HIV replication cycle (e.g. Tenofovir) will likely be effective at blocking the ability of CD123$^+$ ASDCs from becoming infected and transmitting HIV to T cells. However, drugs that block HIV binding to CLRs will likely be required to block CD11c$^+$ ASDCs from performing this function as they transmit the virus independent of infection and replication.

In future, we will use our expertise in high parameter imaging of human tissues [49–51] to carry out imaging mass cytometry and spatial transcriptomics to: i) define ASDCs in all human anogenital tissues [3] and whether they reside in lymphoid aggregates; ii) assess whether they interact or cluster with specific immune cells present in the mucosa and; iii) investigate their relative importance in taking up HIV and transferring it to CD4 T cells in inflamed anogenital mucosae compared to other infiltrating or resident DCs (cDC1, cDC2, DC3 and monocyte derived DCs). This will determine their relative importance in therapeutic strategies that aim at preventing HIV uptake in anogenital mucosae. Given the difference in the biology of pDCs and ASDC subsets and particularly their interactions with HIV, it will be essential to study all three subsets in the pathogenesis of other viruses and diseases. As knowledge progresses, specific interventions related to any of these subsets in other diseases may be become apparent.

## Materials and methods

### Ethics statement

This study was approved by the Western Sydney Local Area Health District (WSLHD) Human Research Ethics Committee (HREC) with reference number (4192) AU RED HREC/15 WMEAD/11.

### Isolation of pDCs and ASDCs from human blood

Peripheral blood mononuclear cells (PBMCs) were isolated via Ficoll-Paque (GE Healthcare Life Sciences, Little Chalfont, United Kingdom) density separation from HIV-seronegative blood supplied by the Australian Red Cross Blood Service, Sydney, Australia. pDCs were isolated from PBMCs using either the Human Plasmacytoid Dendritic Cell Isolation Kit II (Miltenyi Biotec) or by enriching for DCs using the Human Pan DC Enrichment Kit (Miltenyi Biotec). This was followed by cell sorting to isolate pDCs and the two populations of ASDCs: CD123$^+$ and CD11c$^+$. Briefly, cells were resuspended in 150 µl of PBS per 20 x 10$^6$ cells and stained with Live-Dead Near Infra-Red (LDNIR, Invitrogen) for 10 minutes at room temperature and then washed with a fluorescence activated cell sorting (FACS) wash (1% FCS (v/v), 2

mM EDTA, 0.1% sodium azide (w/v) in PBS). Cells were resuspended in 180 μL of Brilliant Stain Buffer and stained for 30 minutes at 4˚C with a combination of different antibodies: CD45 BV786 (HI30, #563716, BD), HLA-DR BUV395 (G46-6, #56404, BD) or BV605 (G46-6, #562845, BD), a Lineage cocktail (Lin1) FITC (#340546, BD) or BV510 (#348807, Biolegend) containing CD3 (SK7, OKT3), CD14 (MφP9, M5E2), CD16 (3G8, 3G8), CD19 (SJ25C1, HIB19), CD20 (L27, 2H7) and CD56 (NCAM16.2, HCD56) antibodies, AXL PE (DS7HAXL, #12-1087-42, Thermofisher), Siglec- APC (REA852, #130-112-711, Miltenyi Biotec), CD123 PE-CF594 (7G3, #562391, BD) or BV421 (7G3, #563362, BD), CD11c PE-Vio770 (REA618, #130-113-588, Miltenyi Biotec clone) or BB15 (B-ly6, #564490, BD), and CX3CR1 BV421 (2A9-1, #565800, BD) and CD141 BB700 (1A4, #742245, BD), and BDCA2 PE-CY7 (201A, #354214, Biolegend). They were then washed in FACS wash and resuspended in RPMI for either phenotyping or cell sorting. For phenotyping, data was recorded on the BD LSR Fortessa using BD FACSDiva (BD Biosciences) and analysed using FlowJo v10.4 (FlowJo LLC). Cell sorting was carried on the BD Influx Cell Sorter using a 100 μm nozzle. pDCs were sorted as live CD45[+] Lin1[-] HLA-DR[+] CD141[-] AXL[-] Siglec-6[-] BDCA2[+] CD123[+] cells. ASDCs were sorted as AXL[+] Siglec-6[+] CD123[+] or AXL[+] Siglec-6[+] CD11c[+] cells. For compensation controls, beads were stained using 1ul of antibody per 1 drop of beads (4˚C for 30 minutes) and washed twice with FACS before acquisition.

## Patients' history

Non-inflamed and inflamed human tissue samples were obtained from patients (S2 and S3 Tables) undergoing colorectal surgery at hospitals in the Westmead Health Precinct, and written consent was collected from all donors. The first rectal tissue was obtained from 10-year-old female (with written consent from the parents) with symptoms of rectal pain and pus discharge and medicated with a hydrocortisone colonic enema. The tissue had features of chronic inflammatory bowel disease and mucosal changes of chronic active colitis. No diagnostic features of Crohn's disease or ulcerative colitis was observed in the rectal stump. The second inflamed human rectal tissue was from a patient suffering from chronic ulcerative colitis. The third sample was a lymph node from an inflamed rectum while the tissue used in Imaging mass cytometry was from an inflamed human colon. Tissues from non-inflamed rectal tissues were extracted from patients during the process of removing nearby cancerous tissues. Inflamed human tissues were collected from abdominal skins from abdominoplasties, labia from labiaplasties and foreskin from circumcisions at Westmead Hospital, Westmead Private Hospital and Hunters Hill Private Hospital in Sydney.

## Enzymatic digestion of non-inflamed and inflamed human tissues to identify ASDCs and pDCs by flow cytometry

To isolate mucosal immune cells from **colorectal** tissues, the lamina propria was mechanically separated from other sublayers and cut into pieces of approximately 5 mm across. 25 mm[2] of tissue was incubated in 20 mL of RF10 with 0.30% DTT (w/v) and 2 mM EDTA for 15 minutes at 37˚C twice to strip the surface mucus and epithelium. The tissue was washed twice in PBS by passing through a tea strainer and digested in 20 mL of RPMI with 0.35% Type 4 Collagenase for 30 minutes at 37˚C twice. The liberated cells were passed through a 100 μm cell strainer twice and washed with DPBS twice. The red blood cell fraction was removed by incubating the cells in 5 mL of 1x Red Cell Lysis Buffer (Biolegend) in sterile $H_2O$ for 3 minutes at room temperature prior to staining for flow cytometry to identify pDCs and ASDCS. Isolated cells were stained with the following antibodies: CD45 BV786 (HI30, #563716, BD), HLA-DR BUV786 (LN3, #417-9956-41, Thermofisher clone), CD3 AF700 (UCHT1, #557943, BD),

CD14 BUV737 (M5E2, #612763, BD), CD16 BUV496 (3G8, #564653, BD), CD19 BV605 (SJ25C1, #562653, BD), AXL PE (DS7HAXL, #12-1087-42, Thermofisher), Siglec-6 APC (REA852, #130-112-711, Miltenyi Biotec), CD123 PE-CF594 (7G3, #562391, BD), and CD11c BV650 (B-ly6, #563404, BD). Data was recorded on the BD LSR Fortessa as above. t-distributed stochastic neighbour embedding (t-SNE) was performed to analyse cell populations using 1000 iterations, a perplexity of 20, a learning rate of 200 and a theta of 0.5, as per the recommended settings.

To isolate immune cells from **abdominal** skin, samples were cut into triangles, stretched out using large surgical forceps and grafted to 1 mm thickness using a skin graft knife. The resulting skin was passed through a skin graft mesher and evenly distributed in 50 ml Falcon tubes to fill approximately 25% of the tube. Dermis and epidermis were separated by incubation in 30ml of filter sterilised RPMI supplemented with 1 U/mL dispase and 0.1% (v/v) gentamicin overnight at 4°C on a rotator. Tubes were placed in a 37°C water bath for 15 minutes to initiate enzymatic activity. The skin was washed in PBS to remove dispase and epidermis was mechanically peeled from dermis using fine forceps. Dermis and epidermis were cut into 5 x 5 mm pieces and placed into separate 50 ml falcon tubes up to the 5 ml mark. Cells were liberated from tissue by an incubation in 20 ml of RPMI supplemented with 200 U/mL Type IV collagenase and 100 U/ml DNase for 120 minutes at 37°C on a rotator. Cells were separated from undigested dermal and epidermal tissues by passing supernatants through a tea strainer twice followed by a 100μm cell strainer into a 50ml falcon tube and washing in PBS (300 x$g$ for 5 minutes). Cells were washed twice more in PBS, ready for antibody staining. For the **genital skin** digestion, any excess underlying connective tissue/fat from each sample was removed using a scalpel and surgical scissors and small cuts were made in the epidermal surface to mimic the action of the skin graft mesher. From herein the tissue was processed similar to abdominal skin, except for the dispase concentration which was increased to 2 U/ml.

After isolating cells from both abdominal and genital tissues, they were stained with Fixable Viability Stain 700 (FVS700) (BD) to identify non-viable cells and a combination of the following antibodies: HLA-DR APC fire810 (L243, #307674, BD) or BUV395 (G46-6, #372608, BD), HLA-DQ BUV563 (TU169, #748563, BD), CD45 BB790 (HI30, #624296, BD), CD3 NovaBlue660-120s (UCHT1, #H002T03B08, Thermofisher), CD19 NovaBlue660-120s (HIB19, #H004T03B08, Thermofisher), CD103 PE Vio615 (REA803, ##130-111-837, Miltenyi Biotec), CD11b BV711 (ICRF44, #301344, Biolegend), CD5 PE Cy5.5 (CD5-5D7, #MHCD0518, Thermofisher), AXL BV785 (108724, #747857, BD), CD123 PE Cy5 (6h6, #306008, Biolegend), Siglec-6 BV480 (767329, #747914, BD), CD11c BUV395 (B-ly6, #563787, BD) or BUV737 (B-ly6, #741827, BD), MR BV750 (19.2, #746891, BD), DC-SIGN BV421 (DCN46, #564127, BD), Siglec-1 BUV615 (7–239, #751124, BD), Langerin PE Vio770 (MB22-9FS, #130-100-586, Miltenyi Biotec), and CD4 BV605 (OKT4, #317437, Biolegend). Data was acquired on the BD FACSymphony using BD DIVA software and analysed by FlowJo (Treestar V 10.9.0). Compensation controls were prepared as previously described in PBMC staining procedure.

## Imaging mass cytometry (IMC)

Inflamed human colon tissue samples were incubated in 4% PFA in PBS for 18–24 hours prior to paraffin embedding and sectioning at 4 μm thickness. Sections for imaging mass cytometry were baked at 60°C for 1 hour, dewaxed in xylene and rehydrated in decreasing concentrations of ethanol from 100% through to 50%. Slides were then washed twice in TBS (0.1% Tween-20) for 2 min. Antigen retrieval was performed using pH 9 Tris-EDTA buffer (10 mM Tris, 1mM EDTA) in a microwave, first at 100% power for 3 min and next at 30% power for 15 min. Slides were then cooled at room temperature for 45 min before being washed in TBS and DPBS for

10 min each. Sections were blocked with Opal Block (Akoya Biosciences) at 37˚C for 45 min and washed twice in PBS for 2 min. Sections were then incubated overnight at 4˚C with the primary antibodies diluted in TBS (0.1% Tween-20) with 1% BSA. All primary antibodies had been previously conjugated to metal isotopes using the Maxpar Antibody Labeling Kits (Fluidigm). The antibodies used were: rabbit CD31 (clone EPR3094, conjugate 155Gd, Abcam), rabbit CD45 (clone D9M8I, conjugate 154Sm, Cell Signaling Technology), mouse E-cadherin (clone 36-E-Cadherin, conjugate 145Nd, BD Biosciences), Goat AXL (polyclonal, conjugate 158Gd, R&D Systems), mouse CD11c (clone 2F1C10, conjugate 167Er, Proteintech), and CD303 (Goat polyclonal, conjugate 175Lu, R&D Systems), which was used to replace CD123 as two CD123 clones did not work in our IMC panel. Slides were washed twice, first in PBS 0.1% Triton-X and next in PBS for 8 min each before being counterstained with Cell-ID Intercalator-Ir (Fluidigm) in PBS for 30 min at room temperature. Slides were then washed in Milli-Q water for 5 min and air dried. Images were acquired using the Hyperion Imaging System (Fluidigm) at a laser power of 5 dB and frequency of 200 Hz.

### Transcriptomic profile of pDCs and ASDCs via NanoString

To investigate the gene expression of chemokines, cytokines, markers involved in antigen presentation and HIV infection, the nCounter pre-designed Human Immunology Panel 2 (NanoString Technologies) was used to quantify and assess the profile of RNA transcripts. Briefly, $10^4$ cells of either pDCs, CD11c$^+$ or CD123$^+$ ASDCs were either mock or HIV infected with HIV-1 BaL at an MOI of 1.5 due to the high expression of HIV restriction factors [52]. At 18 hours post infection (hpi), supernatants were collected and stored at −80˚C while cells were washed in RNAse free PBS and then lysed in 5 µl of 1/3 RNeasy Lysis (RLT) buffer (Qiagen) that was diluted in molecular grade $H_2O$. The 5 µl cell lysate of each cell type was then hybridized with the Reporter Code Set and Capture Probe Set for 24 hours at 65˚C, loaded onto cartridges via the NanoString Prep Station, and processed using the NanoString nCounter digital analyser. Data was normalised and quality control metrics were calculated using the nSolver software package (NanoString Technologies) according to manufacturer's recommendations. Briefly, the data was normalised to 15 housekeeping genes that were included in the NanoString panel. The normalisation factor for all samples ranged between 0.25 and 4.66 which is within the acceptable range of 0.1–10.

### Single cell RNA Sequencing analysis

Publicly available data from E-MTAB-8142 [15], GSE178341 [16] and GSE94820 [4] were downloaded, processed, and analysed in R. Data accessed through E-MTAB-8142 was first sub-setted to only include myeloid cells from non-inflamed and psoriasis lesion tissue. It was then processed using Seurat's reciprocal PCA integration workflow (https://satijalab.org/seurat/). A t-SNE was generated (*RunTSNE*) using dimensions 1:30. Using differential analysis (*FindMarkers*), annotations provided by the authors and known gene signatures, macrophages, cDC1, cDC2, LC, pDC, ASDC, and monocyte clusters were identified. pDCs and ASDCs from data accessed through GSE178341 were identified using annotations provided by the authors (cM07 and cM08, respectively). Additionally, the Seurat function FindTransferAnchors was used to compare data annotations between datasets to assist in verifying annotations (S3 Fig). pDCs and ASDCs were further sub-setted and the processing repeated without integration. Quality control and data processing metrics were described by the original authors.

## Phenotypic profiling of pDCs and ASDCs

To investigate the surface and intracellular protein expression of pDCs, CD11c$^+$ and CD123$^+$ ASDCs, a high parameter 19-colour flow cytometry panel was designed for acquisition on the BD FACSymphony. Blood PBMCs were isolated from whole blood or buffy coats and enriched using the Human Pan DC Enrichment Kit (Miltenyi Biotec). Enriched cells were resuspended in aliquots of 1 x 10$^6$ cells per 100 μL of PBS and non-viable cells excluded using Fixable Viability Stain 700 (FVS700) (BD). Cells were stained with a cocktail of surface antibodies and BD Horizon Brilliant Stain Buffer Plus for 30 min at 4˚C: CD83 APC (HB15e, #551073, BD), FcER1a APC (AER-37 (CRA-1), #334612, Biolegend), CLEC5A APC (283834, #FAB2384A, R&D Systems), XCR1 APC fire 750 (S15046E, #372608, Biolegend), HLA-DR BUV395 (G46-6, #372608, BD), CD5 BUV496 (UCHT2, #741135, BD), HLA-DQ BUV563 (TU169, #748563, BD), CD11c BUV737 (B-ly6, #741827, BD) and BB515 (B-ly6, #564490, BD), CD163 BUV805 (GHI/61, #749201, BD), CD80 PE (L307.4, #557227, BD), CXCR4 PE (12G5, #555974, BD), CLEC4A PE (9E8, #355306, Biolegend), CLEC10A APC (H037G3, #354704, Biolegend), CCR5 PE (REA245, #139-117-356, Miltenyi Biotec), Siglec-1 PE ef610 (7–239, #61-1699-42, Thermofisher), BDCA2 PE-Cy7 (201A, #354214, Biolegend), CD54 FITC (84H10, #IM0726U, Beckman Coulter), CD123 BV421 (7G3, #563362, BD), Siglec-6 BV480 (767329, #747914, BD), Lin BV510 containing CD3 (OKT3), CD14 (M5E2), CD16 (3G8), CD19 (HIB19), CD20 (2H7) and CD56 (HCD56) antibodies (#348807, Biolegend), AXL BV605 (108724, #747861, BD), CD1c BV650 (F10/21 A3, #742749, BD), CD141 BV711 (1A4, #563155, BD), CD86 BV786 2331 (FUN-1), #740990, BD), and CD4 BV785 (OKT4, #317442, Biolegend). For antibodies with identical fluorophore conjugates, drop in panels were used. For SAMHD1, anti-SAMHD1 (I19-18, #MABF933, Merck) was conjugated to APC using the Abcam APC Conjugation Kit (#ab201807, Abcam). For SAMHD1 intracellular staining, cells were first permeabilised with pre-cooled BD Phosflow Perm Buffer II for 2 min at 4˚C in dark. Data was acquired on the BD FACSymphony using BD DIVA software and analysed by FlowJo (Treestar V 10.9.0). Compensation controls were prepared as previously described in PBMC staining procedure.

## Legend plex assay

Culture supernatants were collected from mock and HIV treated pDCs and ASDCs for 18 hours. They were spun at 3000 rpm to remove debris and were then stored at −80˚C until use. Supernatants were diluted 1:1 in RF10 before the cytokine and chemokine levels were measured according to the manufacturer's instructions as per the Human Proinflammatory Chemokine Panel 1 and Human Inflammation Panel 1 (both from BioLegend). Sample data was acquired with the BD FACSCanto II flow cytometer and analysed using BioLegend LEGENDplex data analysis software. The standard range of detection for the inflammatory cytokine was 2–17000 pg/mL while the range for the chemokines was 2–32000 pg/ml.

## T cell proliferation, activation and polarisation

Allogeneic T cells isolated from PBMC using the naïve Pan T cell isolation kit (Miltenyi) were stained with Celltrace Violet (Thermofisher) as per the manufacturer's instruction. For T cell proliferation, FACS sorted pDCs and the two ASDC subsets were added to the Celltrace Violet stained naïve T cells at a ratio of 1 DC:10 T cells in RF10 for 6 days at 37˚C. Cells were then surface stained with Fixable Viability Stain 700 (FVS700) (BD) then surface stained for CD4 BV785 (OKT4, #317442, Biolegend), CD8 BUV737 (SK1, #564629, BD Bioscience), HLA-DR BB515 (G46-6, #564516, BD), CD25 PECy7 (M-A251, #356108, Biolegend), CD11c APC (B-

ly6, #559877, BD), and CD123 PE-Cy5 (6H6, #306008, Biolegend). Cells were acquired on the BD LSRFortessa.

For T cell polarisation, we investigated intracellular transcriptional factors and cytokine production. Unstained naïve T cells were either cultured with pDCs, CD11c$^+$ ASDCs, CD123$^+$ ASDCs, or as T cells alone. Cultures were incubated at 37˚C. On day 6, we assessed intracellular transcription factors by incubating the cells with the BD Pharmingen Transcription Factor Buffer Set (BD bioscience) and antibodies to detect Tbet BV650 (O4-46, #564142, BD), FoxP3 PE Dazzle594 (206D, #320126, Biolegend) and RORγT PE (AFKJS-9, #12-6988-82, Thermo-fisher). For cytokine production, all conditions were stimulated on day 6 with anti-CD3 and anti-CD28 overnight (5µg/mL, Sigma-Aldrich). Supernatants were then collected and assessed for cytokine production by the LEGENDplex Human Th Cytokine Panel (Biolegend). As INF-γ levels were above the standard range of detection via the LEGENDplex, they were assessed by the ELISA MAX Deluxe Set Human (Biolegend).

## HIV infection of pDCs and ASDCs and their viral transfer to CD4 T cells

pDCs, CD11c$^+$ or CD123$^+$ ASDCs (30,000–60,000 cells) were either mock or HIV infected with HIV-1 BaL at an MOI of 1.5 for 2 h at 37˚C. They were then washed twice with RPMI and cultured in 200µl of RF10 in 96 U well bottom shaped plates. JLTR T cells were added to pDCs and ASDCs at the ratio of 1 DC: 3 JLTRs at 2 and 96 hpi. On day 4 post co-culture, cells were fixed in 4% PFA then HIV transfer to JLTR cells was determined by flow cytometry by measuring GFP expression, as JLTRs express GFP under the control of the HIV-1 promotor. Half the supernatants were collected prior to the addition of JLTR at 96 hpi to assess infectious virus release from pDCs and ASDCs. TZM-BL cells (NIH AIDS Research and Reference Reagent Program, contributed by John Kappes and Xiaoyun Wu) were exposed to supernatants and then the LTR β-galactosidase reporter gene expression was measured after a single round of infection. Furthermore, to show that second phase transfer of HIV from ASDCs to CD4 T cells was dependent on their productive infection, we either mock or pre-treated ASDCs with the CCR5 inhibitor, Maraviroc (10µM), for 1 h prior to HIV infection. Cells were cultured for 18 h then washed twice to remove residual virus and Maraviroc. They were cultured for 96 h before the addition of JLTRs. After 5 days of co-culture, GFP expression in JLTRs was assessed by flow cytometry.

## Quantification and Statistical analysis

For comparisons between more than two groups, a repeated measure one-way analysis of variance (ANOVA) with Tukey's post-hoc test was conducted. For HIV viral transfer, statistical comparisons between two groups were performed using Wilcoxon signed-rank tests. All statistical analysis was performed in GraphPad Prism 8, and $p < 0.05$ was considered statistically significant for all tests; $^*p < 0.05$, $^{**}p < 0.01$, $^{***}p < 0.001$, $^{****}p < 0.0001$. Error bars represent standard error mean across all analyses.

## Supporting information

**S1 Fig. Identification of ASDCs and pDCs isolated using the Human Plasmacytoid Dendritic Cell Isolation Kit II.** t-stochastic distributed neighbour embedding (t-SNE) analysis performed on live single Lin1-HLA-DR$^+$ cells based on AXL, Siglec-6, CD123, and CD11c. **(a)** Representative t-SNE dot plot shows the distribution of AXL$^+$ Siglec-6$^+$ DCs (orange) and AXL$^-$ Siglec-6$^-$ CD123$^+$ pDCs (blue) on t-SNE plot. **(b)** Heat map visualisations of the median fluorescence intensity of surface AXL, Siglec-6, CD123, and CD11c expression for populations

on t-SNE plot.
(TIF)

**S2 Fig. Transcriptional profiling of pDCs and ASDCs for gene expression of chemokine receptors from publicly available scRNAseq dataset.** Sorted blood PBMCs were transcriptionally profiled by scRNAseq (GSE94820)[4]. pDC, CD11c[+] ASDC and CD123[+] ASDC annotations were determined using metadata provided by the authors. Chemokine receptors gene expression is shown in pDCs, CD123[+] and CD11c[+] ASDCs.
(TIF)

**S3 Fig. Protein expression of lectin receptors on blood ASDCs and pDCs.** Pan DCs were FACS sorted into CD11c[+] ASDCs, CD123[+] ASDCs and pDCs and surface stained for flow cytometry. Representative plots show the percentage expression of Langerin, MR and DC-SIGN displayed as an FMO (grey) overlayed with true expression for each population. As the percent of positive population was small, expression was displayed as a percentage rather than gMFI.
(TIF)

**S4 Fig. (a)** Pelka et al. 2021 data was downloaded from the GEO (GSE178341). Myeloid clusters (cM01-10) were isolated and analysed separately. The tSNE coordinates used by the authors were downloaded from the Broad Institute's Single Cell Portal (SCP1162). ASDCs and pDCs described by the authors were isolated and reclustered for analysis. **(b-c)** Reynolds et al. 2021 data was downloaded from Developmental Human Cell Atlas (E-MTAB-8142). Myeloid cells, as annotated by the authors, were isolated and analysed separately. Data was batch corrected using Seurat's RPCA method, and clustered using a resolution of 0.6, resulting in 22 clusters. Seurat's *FindTransferAnchors* and *TransferData* functions were used to determine predicted annotations, with GSE178341 (or colorectal cancer) acting as the reference data to E-MTAB-8142 (or psoriasis skin) as the query. A DotPlot was used to demonstrate the prediction scores generated by the functions. ASDCs and pDCs, identified as cluster 15, were isolated and reclustered for analysis.
(TIF)

**S5 Fig. Flow cytometry gating strategy identifying pDCs, ASDCs and cDC2s in blood.** Blood pan DCs were isolated and stained for flow cytometry using a modified panel to allow for the identification of cDC2s. All cells were first gated as Live CD45[+] HLA-DR[+] CD3[-] CD19[-]. Subsequent cell populations were identified as CD123[+] ASDCS (CD11b[-] CD16[-] AXL[+] Siglec-6[+] CD123[+]), CD11c[+] ASDCs (CD11b[-] CD16[-] AXL[+] Siglec-6[+] CD11c[+]), pDCs (CD11b[-] CD16[-] AXL[-] Siglec-6[-] CD11c[-] CD123[+]) and cDC2s (CD11b[-] CD16[-] AXL[-] Siglec-6[-] CD123[-] CD11c[+] CD1c[+] XCR1[-] CD163[-]).
(TIF)

**S6 Fig. T cell proliferation, activation and polarisation after co-cultured with pDCs and ASDCs.** FACS sorted pDCs, CD123[+] and CD11c[+] ASDC were cultured for 7 days at 37°C with Cell trace Violet-stained naïve T cells at a ratio of 1 ASDC or pDC: 10 T cells. Cultures were analysed by flow cytometry to assess **(a)** CD8+ T cell proliferation, **(b)** CD8+ T cell activation via CD25 expression, **(c)** CD4+ T cells percentage expression of transcription factors Tbet, FoxP3 and RORγT. Data presented as mean of ±SD. For all data, *p $< 0.05$ using one-way ANOVA with Tukey's multiple comparisons test.
(TIF)

**S7 Fig. Gene profiling of pDCs and ASDCs in mock and HIV conditions.** pDCs and ASDCs were profiled using NanoString as either mock or after 18 hours of HIV exposure. **(a)** Markers

defining pDCs, CD11c$^+$ and CD123$^+$ ASDCs; **(b)** Chemokines (CCL3, 4, 5; CXCL10 and 8), **(c)** cytokines (TNFα, IL-1B), **(d)** IFN, **(e)** ISGs in mock and HIV exposed pDCs and ASDCs. Data presented as mean of ±SD. For all data, *p < 0.05, **p < 0.01, ***p < 0.001, ****p < 0.0001 using one-way ANOVA with Tukey's multiple comparisons test.
(TIF)

**S1 Table. Nanostring gene list for ASDCs and pDCs.** The gene list detected by NanoString in mock and HIV exposed pDCs, CD11c and CD123 ASDCs. Tab 1 shows the normalised NanoString data and tab 2 showed the gene list that identified pDCs and the two subsets of ASDCs.
(XLSX)

**S2 Table. Patient information from inflamed tissue samples Fig 4.** Information on tissues collected from patients, their medical condition, age/sex (when disclosed) and reason of tissue removal used in Fig 4a–4g to identify ASDCs and pDCs in inflamed tissues.
(TIF)

**S3 Table. Patient information from inflamed tissue samples Fig 5 and relative proportions of pDCs, ASDCs and cDC2.** Information on tissues collected from patients, their medical condition, age/sex (when disclosed), reason of human genital tissues removal, and the relative proportions of pDCs, ASDCs and cDC2 as a percentage of live CD45$^+$ HLA-DR$^+$ CD3$^-$ CD19$^-$ cells used in Fig 5 to identify pDCs, ASDCs and cDC2 in inflamed human skin and genital tissues: abdominal epidermis (n = 1), labia epidermis (n = 2), inner foreskin epidermis (n = 1), labia dermis (n = 3), outer foreskin dermis (n = 1) and inner foreskin lamina propria (n = 1).
(TIF)

**S1 Data.** This file contains the source data used to generate all graphs in this manuscript. Each tab of the Excel file contains the data for a separate figure and data is grouped into panels as presented in the figures.
(XLSX)

## Acknowledgments

Flow cytometry and cell sorting was performed in the Flow Cytometry Core Facility that is supported by The Westmead Institute for Medical Research, Westmead Research Hub, Cancer Institute New South Wales and National Health and Medical Research Council (NHMRC), Australia. NanoString was performed in the genomic Core Facility that is supported by The Westmead Institute for Medical Research and Westmead Research Hub.

## Author Contributions

**Conceptualization:** Orion Tong, Najla Nasr.

**Data curation:** Freja A. Warner van Dijk, Orion Tong, Thomas R. O'Neil.

**Formal analysis:** Freja A. Warner van Dijk, Orion Tong, Kate Jenns, Najla Nasr.

**Funding acquisition:** Anthony L. Cunningham, Najla Nasr.

**Investigation:** Freja A. Warner van Dijk, Orion Tong, Kevin Hu, Heeva Baharlou, Kate Jenns, Najla Nasr.

**Methodology:** Freja A. Warner van Dijk, Orion Tong, Kirstie M. Bertram, Erica E. Vine, Najla Nasr.

**Project administration:** Najla Nasr.

**Resources:** Martijn P. Gosselink, James W. Toh, Tim Papadopoulos, Laith Barnouti, Gregory J. Jenkins, Gavin Sandercoe, Kerrie J. Sandgren.

**Supervision:** Kirstie M. Bertram, Anthony L. Cunningham, Najla Nasr.

**Validation:** Kirstie M. Bertram, Najla Nasr.

**Visualization:** Freja A. Warner van Dijk, Orion Tong, Thomas R. O'Neil, Najla Nasr.

**Writing – original draft:** Freja A. Warner van Dijk, Thomas R. O'Neil, Najla Nasr.

**Writing – review & editing:** Freja A. Warner van Dijk, Kirstie M. Bertram, Erica E. Vine, Muzlifah Haniffa, Kerrie J. Sandgren, Andrew N. Harman, Anthony L. Cunningham, Najla Nasr.

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
