## [Decision Letter · Decision Letter 0]

1 Mar 2024

Dear Dr Nasr,

Thank you very much for submitting your manuscript "Characterising plasmacytoid and myeloid AXL+ SIGLEC6+ dendritic cell functions and their interactions with HIV." for consideration at PLOS Pathogens. As with all papers reviewed by the journal, your manuscript was reviewed by members of the editorial board and by several independent reviewers. In light of the reviews (below this email), we would like to invite the resubmission of a significantly-revised version that takes into account the reviewers' comments.

We cannot make any decision about publication until we have seen the revised manuscript and your response to the reviewers' comments. Your revised manuscript is also likely to be sent to reviewers for further evaluation.

Sincerely,

Daniel C. Douek

Academic Editor

PLOS Pathogens

Richard Koup

Section Editor

PLOS Pathogens

Michael Malim

Editor-in-Chief

PLOS Pathogens

orcid.org/0000-0002-7699-2064

Reviewer's Responses to Questions

**Part I - Summary**

Reviewer #1: In this work, the authors have analyzed the Axl+ Siglec6+ dendritic cell subset (AS DC) in human blood and anogenital tissues, and assessed their interaction with HIV virus in ex vivo assays. A particular attention was paid to the 2 subpopulations of AS DC (CD11c high or CD123 high).

Main claims are

1. AS DC are present in inflamed tissues in human

2. AS DC are more efficient than pDC for stimulating T cells and for inducing cytokine polarization

3. Both subpopulations of AS DC can transmit HIV virus using different mechanisms

Although the experiments reported in the manuscript are well performed, there are a number of points that need to be addressed. In addition, the novelty of claims 1&2 is compromised by previous publications (which are not properly referenced in the manuscript).

Reviewer #2: In this manuscript, Warner van Dijk et al. describe the tissue distribution and function of a recently described dendritic cell population known as Axl+Siglec-6+ DC (ASDC). This work is highly relevant as ASDC has been previously shown to be susceptible to HIV-1 infection and able to transfer virions to T cells (PMID: 31591213). Yet it remains critical to define their role in initial HIV-1 infection.

The authors in this study describe two population of Axl+Siglec-6+ DC (ASDC) distinguished by the expression of CD11c and CD123. They compare these 2 populations, along with pDC (CD123+ Axl- DC) in term of phenotype, presence in peripheral tissues (intestinal tract and skin), ability to promote T cell activation and polarization, susceptibility to HIV-1 infection, and capacity to transfer and transmit the virus to CD4+ T cells. The authors used bulk RNA sequencing (Nanostring) of sorted population from healthy donor blood to define the signatures associated with the two ASDC populations. These signatures were then compared to publicly available data sets from tissues (skin, colon). Immunophenotyping by flow cytometry was performed in parallel in blood and cells from peripheral tissues obtained from a limited number of donors. The authors conclude that the increased levels of ASDC in inflamed tissues could be central in HIV-1 transmission (which occurs mainly through anogenital infection and is favoured by inflammation).

Due to the limited cell numbers extracted from tissues, all functional assays (T cell activation and polarization, HIV-1 infection) were performed using peripheral blood ASDCs. Consequently, some conclusions are not supported by direct evidence. While these results are of interest for the scientific community, important questions must be addressed to validate the significance of the findings presented and their relevance. Most of my numerous comments call for text editing or additions and reanalysis of the data rather than performing new experiments.

**Part II – Major Issues: Key Experiments Required for Acceptance**

Reviewer #1: Main issues

1. The authors claim that AS DC have only been described in blood and tonsils. However, there are several articles showing the presence of bona fide AS DC in spleen (10.1126/science.aag300), bone marrow (10.1126/science.aag300; 10.1016/j.immuni.2017.11.001 ), inflamed cerebrospinal fluid (10.1016/j.clim.2023.109686 ), inflamed broncho-alveolar lavage (10.1038/s41467-019-09913-4), inflamed skin (10.1084/jem.20190811). References need to be updated. The claim ‘we have shown for the first time that AS DC are present in inflamed human tissues’ (line 175) needs to be revised and tuned down.

2. The analysis shown in figure 3 needs to be better explained. Why did the authors subset the data on monocytes (line 631) ? The different subsetting steps should be shown (at least in supplementary data) to make this analysis more convincing. In addition, UMAP has now become the standard representation for this type of analysis instead of t-SNE.

3. The superior ability of AS DC to stimulate T cell proliferation compared to pDC, and the higher ability of CD11c+ AS DC has been shown previously (10.1016/j.immuni.2017.11.001). This needs to be mentioned.

4. Although the reviewer understands that obtaining human tissues is difficult, it is problematic to draw conclusions from data derived from only one biological sample (figure 8). It is also very difficult to compare this data to blood AS DC and pDC, because histograms are not shown in figure 8. As it stands, the data reported in figure 8 seems rather preliminary.

Reviewer #2: 1 The authors assessed the ability of the ASDCs and pDCs to polarize T helper (Th) cells (Figure 5c). Th polarization was evaluated based on the cytokines present in the culture supernatant after a 6-day incubation of T cells with antiCD3/CD28 +/- DC subsets. The authors conclude from these results that Th1 are not induced (line 40, 223, 380). However, Interferon gamma, which is the canonical Th1 cytokine, was not measured. To support authors’ conclusions, Th polarization should be confirmed by analyzing transcription factors and performing intracellular cytokine staining.

Throughout the manuscript, data are presented with SEM, which quantifies uncertainty in estimate of the mean. Standard deviation (SD) indicates dispersion of the data from mean (PMID: 23125963). To visualize the variability within sample, descriptive data should be precisely summarized with SD instead of the SEM in Figure 1, 4h, 5-7.

The authors utilize the co-expression of Axl and Siglec-6 to define ASDCs similarly to the approach by Villani et al. (PMID: 28428369). Through Tsne representation of their flow cytometry data, they show that ASDCs can be segregated into 2 clusters with differential expression of CD123 and CD11c.

Since the authors claim an enrichment of CD11c+ ASDCs in inflamed tissues, it is important to exclude that those cells are distinct from bona fide CD1c+ DC2. This distinction is crucial to substantiate the authors' conclusions, particularly considering the findings in Figure 2, which indicate that CD11c+ ASDCs express numerous co-stimulatory molecules (lines 135-136), suggestive of activated DC2 cells.

How do the 2 ASDC populations described in the manuscript correspond to the ASDCs previously identified by single-cell RNAseq? Please compare the 2 ASDC subsets to both DC2 and DC5 (PMID: 28428369) and pre-DC (PMID: 28473638).

**Part III – Minor Issues: Editorial and Data Presentation Modifications**

Reviewer #1: It is not always clear in the text when AS DC purified from blood are used (versus anogenital tissue). This needs to be clarified.

The discussion part is too long.

Reviewer #2: - Please provide the gene list that was used to investgate ASDC enrichment in tissues (Figure 2b and 3).

- What is the status of DC2 cells in healthy and inflamed tissues? Please incorporate DC2 into your analyses to differentiate CD11+c ASDC from DC2.

- The expression of CD11c has been documented for CD123+ ASDCs (PMID: 31825848; 36866043). How does this population correlate with the CD11c+ ASDCs presented in this study?

1. RNA vs. protein expression

The authors present results from RNA sequencing data using Nanostring technology (Figure 2 and 4h), and protein expression by flow cytometry (Figure 7d) in different figures, making it challenging for readers to follow. Furthermore, the data presented are not always consistent with previous results used for comparison (Figure 7d vs. S6, PMID: 28428369).

- In Fig 5., RNA and Protein expression do not consistently align, as seen with the markers analyzed in Fig 5a compared to the Nanostring results in Fig2. How do the authors account these discrepancies?

- In Fig 4h. CD123+ ASDCs exhibit the lowest RNA expression of CCR5 and CXCR4. How does these results compare with previous results? How can the reported susceptibility to HIV-1 occur without co-receptor expression?

2. ASDCs in tissues

- Apart from quantifying ASDCs in inflamed tissues using publicly available transcriptomic datasets, the authors enrolled additional patients to investigate these cells through flow cytometry and imaging mass cytometry. The results are depicted in Figure 4 and S2.

- Please provide patient information: age, sex, underlying condition, tissue collected.

- In Fig 4e, an FMO for Siglec-6 should be included to ensure proper gating. Why does the ASDC gate for this sample differ from all the others (Siglec-6 MFI > 103)?

- The gating strategy in Figure 4 lacks consistency. Although different antibody panels are used in different experiments, they all include similar markers to define the 2 populations of ASDC: Axl, Siglec-6, CD123 and CD11c (Figure 1 to 4f). However, for the inner foreskin sample depicted in Figure 4g, CD11b and CD5 markers are also used.

o Please explain why a different gating strategy is used?

o Is this gating strategy similar to the one used to quantify ASDC in Figure S2?

- ASDCs have been previously reported in tonsils and LN, but their presence in mucosal tissues remains unknown. In addition to flow cytometry, authors utilized imaging mass cytometry to confirm the presence of ASDC in inflamed colon (Figure 4f) and discovered that ASDC are located at the periphery of a lymphoid aggregates suggestive of a tertiary lymphoid structure.

o Can the authors clarify whether they believe lymphoid aggregates are essential for the presence of ASDC presence in tissue?

o The authors write (line 528) “the tissue used in Imaging mass cytometry were from two inflamed human colons” but only 1 sample is displayed in Figure 4f. Please include the data of the second sample or modify the material and method accordingly.

- The authors mentioned that they“did not observe pDCs or ASDCs in non-inflamed rectal tissue, confirming their presence in inflamed tissues only (Fig 4)” (line 365). However, ASDCs were also detected in non-inflamed skin tissues (Figure 4g and S2), including foreskin. Could the authors provide insight into this discrepancy?

3. Th polarization

I am intrigued by the little variations observed in the MLR assays illustrated in the histograms of Fig 5b and c. Were these experiments conducted with distinct donors, or are they technical replicates? Despite the robust proliferation observed with both types of ASDCs, the levels of IL2 quantified in the supernatant (Fig 5d) appear notably low in comparison.

4. HIV infection

The phase termed the "1st phase of infection," involving a 2-hour exposure to HIV followed by washes and coculture with reporter T cells (Fig 7a), cannot be interpreted as indicative of different transmission modes utilized by the three cell populations tested. The histogram presented with n=3 indicates nonsignificant differences. Given that pDCs do not become infected by HIV, the observed 3.5% transmission likely represents background noise in their assay.

The "second phase of transmission" reveals minimal distinctions between cell types, with the difference between the two ASDC types not reaching statistical significance. This observation is consistent with the findings from the TZMBL transfer assay (Fig 7b).

The authors noted that “all characteristic genes for blood pDCs and ASDCs identified in the heat map of (Fig 2b) were downregulated upon HIV exposure” (line 231). In Figure S5 “Number of RNA counts” are displayed. How can authors rule out that variations in RNA counts result from differential susceptibility of the various DCs to cell death in their in vitro culture (18h pi)? Can the data be normalized to housekeeping genes? Were the cells counted before performing Nanostring?

5. The abstract

The sentence “This indicates that the previously observed T cell stimulatory effects of pDCs are due to contaminating CD123 ASDCs” is inappropriate as this has already been proposed in previous studies.

The phrase “slightly more expressed on CD11c ASDCs” is imprecise and inappropriate. The authors solely assessed expression at the RNA level by nanostring (Fig 2 b, heatmap). This should have been checked at the protein level and quantified.

Due to the aforementioned reasons, the authors cannot conclude that “ pDCs, CD11c+ and CD123+ ASDCs expressed unique repertoires of HIV binding receptors and transmitted HIV to CD4 T cells via different modes”.

Minor points:

Figure 5. Please verify the labelling of the different groups as discrepancies exist between the dotplots and the summative bar plots.

- Figure 6. The figure legend should specify the strain of HIV-1 used in these experiments.

- Line 203. CD83 was found to be expressed by CD11

---

## [Decision Letter · Decision Letter 1]

18 May 2024

Dear Dr Nasr,

Thank you very much for submitting your manuscript "Characterising plasmacytoid and myeloid AXL+ SIGLEC-6+ dendritic cell functions and their interactions with HIV." for consideration at PLOS Pathogens. As with all papers reviewed by the journal, your manuscript was reviewed by members of the editorial board and by several independent reviewers. The reviewers appreciated the attention to an important topic. Based on the reviews, we are likely to accept this manuscript for publication, providing that you modify the manuscript according to the review recommendations.

Sincerely,

Daniel C. Douek

Academic Editor

PLOS Pathogens

Richard Koup

Section Editor

PLOS Pathogens

Michael Malim

Editor-in-Chief

PLOS Pathogens

orcid.org/0000-0002-7699-2064

Reviewer Comments (if any, and for reference):

Reviewer's Responses to Questions

**Part I - Summary**

Reviewer #1: In the revised version of their manuscript, the authors have adequately addressed the reviewers' comments.

Reviewer #2: The referee acknowledges again the important work performed on cell populations very difficult to obtain. However, the authors have only partially addressed the critics and requests previously made.

**Part II – Major Issues: Key Experiments Required for Acceptance**

Reviewer #1: In the revised version of their manuscript, the authors have adequately addressed the reviewers' comments.

Reviewer #2: Page 4. On the comparison of CD123+ and CD11c+ ASDC gene signature with previous RNAseq data sets.

The authors comprehensively and thoroughly compared the phenotype of ASDC populations with previous articles. However, it is also important to validate the gene signature obtained for the 2 ASDC populations described with known data sets. Especially as the gene signature the authors used (Fig 2b, Table S1) are issued from ex vivo DC after culture (Mock- or HIV-culture), which could impact greatly gene expression.

About SAMHD1:

DCs are thought to express high levels of SAMHD1, which render them resistant to HIV-1 infection. Previous reports also show that addition of Vpx to HIV-1 particles that induces SAMHD1 degradation leads to 10-time increase of Axl+Siglec6+ DC infection rate (PMID: 31591213). In Figure 2e, the authors report a low expression of SAMHD1 protein (2-7%). I acknowledge the difficulty to perform intracellular staining with very few cells. However, as the authors emphasize a differential susceptibility to HIV-1 infection of the 2 ASDC populations and relate it to SAMHD-1 expression, it would be important to show positive control of SAMHD1 staining (e.g. on THP1 cells) and validate the results maybe by pooling different donors.

About cDC2. Page 3.

Figure 6A from PMID 28428369 represent peripheral DCs at steady state. To exclude any overlap between cDC2 and ASDC, it is important to show that ASDCs do not express CD1c. The gating strategy used in Figures 1, 4 and S5 do not allow to verify this particular point of importance.

**Part III – Minor Issues: Editorial and Data Presentation Modifications**

Reviewer #1: In the revised version of their manuscript, the authors have adequately addressed the reviewers' comments.

Reviewer #2: (No Response)

PLOS authors have the option to publish the peer review history of their article (what does this mean?). If published, this will include your full peer review and any attached files.

Reviewer #1: No

Reviewer #2: No

Figure Files:

Data Requirements:

Reproducibility:

References:

---

## [Decision Letter · Decision Letter 2]

18 Jun 2024

Dear Dr Nasr,

We are pleased to inform you that your manuscript 'Characterising plasmacytoid and myeloid AXL+ SIGLEC-6+ dendritic cell functions and their interactions with HIV.' has been provisionally accepted for publication in PLOS Pathogens.

Best regards,

Daniel C. Douek

Academic Editor

PLOS Pathogens

Richard Koup

Section Editor

PLOS Pathogens

Michael Malim

Editor-in-Chief

PLOS Pathogens

orcid.org/0000-0002-7699-2064

Reviewer Comments (if any, and for reference):

Reviewer's Responses to Questions

**Part I - Summary**

Reviewer #2: The authors have addressed my concerns

**Part II – Major Issues: Key Experiments Required for Acceptance**

Reviewer #2: NA

**Part III – Minor Issues: Editorial and Data Presentation Modifications**

Reviewer #2: NA

PLOS authors have the option to publish the peer review history of their article (what does this mean?). If published, this will include your full peer review and any attached files.

Reviewer #2: No

---

## [Editor Report · Acceptance letter]

22 Jun 2024

Dear Dr Nasr,

We are delighted to inform you that your manuscript, "Characterising plasmacytoid and myeloid AXL+ SIGLEC-6+ dendritic cell functions and their interactions with HIV.," has been formally accepted for publication in PLOS Pathogens.

Best regards,

Michael Malim

Editor-in-Chief

PLOS Pathogens

orcid.org/0000-0002-7699-2064